# Local genetic sex differences in quantitative traits

Emil Uffelmann [1] ✉, Christiaan de Leeuw [1], Marijn Schipper [1] & Danielle Posthuma[1,2]

Many traits show small global sex differences in genetic correlations and heritability. However, how these differences are distributed across the genome remains unknown. Here, we use LAVA to test for local genetic sex differences in genetic correlations, heritabilities, and the magnitude of genetic effects across 157 quantitative traits in the UK Biobank. Nearly every trait shows evidence for sex-dimorphic effects in at least one locus. We find that such loci can flag biological differences between the sexes. Moreover, we test for differences in the magnitude of genetic effects on the raw and the standardized scale. We show these have complementary interpretations, where only the latter scale is informative for heritability. Our results show how average metrics of genetic correlation and heritability across the whole genome can mask important variability between loci and that the scale of genetic effects needs to be considered carefully when comparing their magnitudes.

Males and females differ in prevalence, severity, and age of onset across a wide range of diseases[1,2]. Similarly, many quantitative traits are sexually dimorphic in their means and variances[3,4]. These observed sex differences may partly be due to differences in the external environment (e.g., societal gender norms[5,6]) or the cellular environment (e.g., hormone and gene expression levels[7,8]). As such, sex may be regarded as an environmental variable that interacts with genetic variants (GxS) to produce partially dimorphic phenotypes[3].

Male and female autosomal allele frequencies do not differ due to Mendel's law of segregation, which is independent of the sex chromosomes[9]. Therefore, genetic sex differences are expected to manifest biologically only in causal variant effect sizes. Importantly, the effect size scales must be considered when comparing males and females because they can have different interpretations. For example, sex-divergent heritability estimates can only occur if effect sizes differ on the standardized scale.

Genome-wide association studies (GWASs) typically analyze males and females together while including sex as a covariate, thereby potentially masking sex-specific genetic effects. Recently, large-scale sex-stratified GWASs have been conducted that compare global heritabilities ($h^2_{\text{global}}$) and estimate global genetic correlations ($r_{\text{g, global}}$)

between males and females[4,10–13]. These studies have identified many traits that show genetic sex differences on a global level. However, such global analyses cannot elucidate the local architecture of genetic sex differences[14]. Commonly, studies rely on comparing the presence and absence of genome-wide significant loci between males and females[15,16]. However, a locus that is just below the significance threshold for one sex and hovers just above the threshold for the other may not point to a meaningful difference between the sexes. This is because the difference between "significant" and "not significant" may not itself be significant[17].

Furthermore, a recent study suggests that GxS mainly acts through differences in the magnitude of effect sizes between males and females and not in the identity of causal variants[18]. While this marks an important advance in the study of GxS, the magnitude of effect sizes was only considered on the raw scale. The scale of effect sizes needs to be considered because it can have different yet equally important interpretations. For example, only the standardized scale is informative for the heritability.

To address these gaps and gain insight into local (dis-)similarities in genetic signals between males and females, we estimate local genetic correlations and test whether these differed from one,

[1]Department of Complex Trait Genetics, Center for Neurogenomics and Cognitive Research, Amsterdam Neuroscience, Vrije Universiteit Amsterdam, Amsterdam, The Netherlands. [2]Department of Child and Adolescent Psychiatry and Pediatric Psychology, Section Complex Trait Genetics, Amsterdam Neuroscience, Vrije Universiteit Medical Center, Amsterdam University Medical Center, Amsterdam, The Netherlands. ✉e-mail: e.uffelmann@vu.nl

compare local heritability estimates, and test for equality of genetic effects on a regional (i.e., 1 Mb regions) and a gene level in 157 quantitative traits. While the vast majority of loci do not significantly differ between males and females, almost every trait we study has at least one locus that does. We show that these loci can highlight trait-relevant biology. Lastly, we evaluate the equality of local genetic effects on the raw phenotypic, as well as on the standardized scale. While the results from both scales correlate strongly, discrepancies exist that can be informative in interpreting observed sex differences.

## Results

### Data overview

Sex-stratified GWAS summary statistics were downloaded from the Nealelab (https://github.com/Nealelab/UK_Biobank_GWAS); sex was genetically inferred[19]. We analyzed 157 quantitative traits across 13 ICD/ICF domains with a combined maximum sample size of 360,564 (sample size varies across traits; see Supplementary Data 1). Subjects were of British ancestry (see "Methods" for details).

### Local heritability

The autosomal genome was divided into 2495 semi Linkage Disequilibrium (LD)-independent loci of approximately 1 Mb in size (see "Methods" and ref. [20]). We computed local heritability estimates ($h^2_{local}$) for all traits with LAVA[20]. We found 146 loci across 47 traits to significantly differ between males and females at a Bonferroni-corrected significance threshold of $p < 0.05/(2495 \times 157) = 1.28e\text{-}07$ (see Fig. 1, "Methods" for a description of the test of equal heritabilities, and Supplementary Data 2 for type-1 error simulation results). 55% of these loci (80/146) had larger $h^2_{local}$s in females, which was not significantly different from 50% ($p = 0.28$). Across all traits, $h^2_{local}$s for loci that were significant in both males and females correlated strongly ($r_{pearson} = 0.98$, $p < 1.00e\text{-}300$). After the exclusion of loci with very large $h^2_{local}$s (i.e., > 0.2, for lipoprotein A, direct bilirubin, and total bilirubin; see Supplementary Fig. 1), the correlation estimate decreased but remained high ($r_{pearson} = 0.89$, $p < 1.00e\text{-}300$).

Approximately 62% of all significant $h^2_{local}$s are significant in only one of the sexes. Across all traits, we identify more loci with significant $h^2_{local}$s in females than males (10614 loci across 153 traits vs. 8767 loci

across 151 traits). This is at least partly due to the sex-stratified GWASs' sample sizes being mostly larger in females, increasing the power of $h^2_{local}$ analyses in females relative to males. Specifically, 153 out of 157 traits have larger sample sizes in females with a median female:male sample size ratio of 1.16 (see Supplementary Fig. 2 and Supplementary Data 1 for exact sample sizes). This is a consequence of the female sampling bias in the UK Biobank. Simulations show that this can result in power asymmetries at large sample size differences (see Supplementary Fig. 3 and "Methods"), with somewhat greater power to detect differences in heritability if the larger heritability is for the sex with the smaller sample rather than the other way around. However, this power asymmetry is largely negligible for the sample size differences typical of the UK Biobank.

Mostly blood biomarker traits show strong $h^2_{local}$ differences in magnitude for individual loci. For example, locus 2181 (17:7264459:8554763) for testosterone has a $h^2_{local}$ of 4.3% ($p < 1.00e\text{-}300$) for males but 0.06% ($p = 3.79e\text{-}05$) for females. Locus 963 for Rheumatoid factor (6:32454578:32539567) has a $h^2_{local}$ of 1% ($p = 3.19e\text{-}19$) for females, but 0.02% ($p = 0.4$) for males. For Urate, four neighboring loci (4:8882617:11050119) have significant and large $h^2_{local}$s for both males and females, but the female estimates are 4–5 times larger (female range: 0.5%–10%; male range: 0.1%–2.7%). SLC2A9, a Urate solute carrier and main GWAS hit[16], is located within these loci, which suggests sex differences in the relative importance of this gene for Urate blood levels.

### Local genetic correlations

To test for local genetic sex differences, we computed local genetic correlations ($r_{g,\ local}$) with LAVA for every trait and locus with sufficient genetic signal (i.e., we used a $h^2_{local}$ $p$-value threshold of $p < 1.00e\text{-}04$; see "Methods") for both males and females ($N_{loci} = 11259$) and tested if the $r_{g,\ local}$ is significantly different from one (i.e., we tested for deviation from perfect correlation) at a Bonferroni-corrected significance threshold of $p < 0.05/11259 = 4.44e\text{-}06$. We found 118 traits with at least one locus where the $r_{g,\ local}$ is significantly different from one (see Fig. 2 and Supplementary Fig. 4). Moreover, we found 205 loci across 103 traits to be significantly different from one and with negative $r_{g,\ local}$. Again, we found blood biomarker traits (metabolic and immunological traits) exhibited the largest number of differences between the sexes. IGF-1 and testosterone had the largest number of loci whose correlation was significantly different from one, namely 12 and 11, respectively (Supplementary Fig. 4). For some traits, such as urate, total and direct bilirubin, and lipoprotein A, we found loci where the $r_{g,\ local}$ was significantly different from one while being very close to one (see Fig. 2). These loci have large $h^2_{local}$s in both males and females, thus having more precise $r_{g,\ local}$ estimates, and higher statistical power to detect subtle deviations from one. We note that 8% (942/11,935) and 7% (842/12,246) of cross-trait pairs have significant Bonferroni-corrected global genetic correlations ($r_{g,\ global}$) for females and males, respectively (see Supplementary Figs. 5 and 6). These mostly cluster within weight- and fat-distribution-related phenotypes, and some loci with significant $r_{g,\ local}$ between males and females are expected to repeat across these phenotypes. Indeed, locus 2310 (19:3085447:3893909) has a significant $r_{g,\ local}$ for 14 metabolic traits. However, this is not the norm, and 80% of loci with at least one significant $r_{g,\ local}$ are found for at most two traits.

In addition to local genetic correlations, we computed global genetic correlations ($r_{g,\ global}$) using LD Score Regression (LDSC)[21] and compared them to the weighted mean of $r_{g,\ local}$s across all loci for which $r_{g,\ local}$s could be computed using LAVA (see Fig. 3 and "Methods"). While 20.13% (30/149) of traits had LDSC-$r_{g,\ global}$s significantly different from one, most were very close to one, with the notable exception of testosterone. To ensure some reliability of LAVA mean $r_{g,\ local}$s, we only considered traits for which at least ten $r_{g,\ local}$s could be computed. Overall, both methods have good agreement (mean

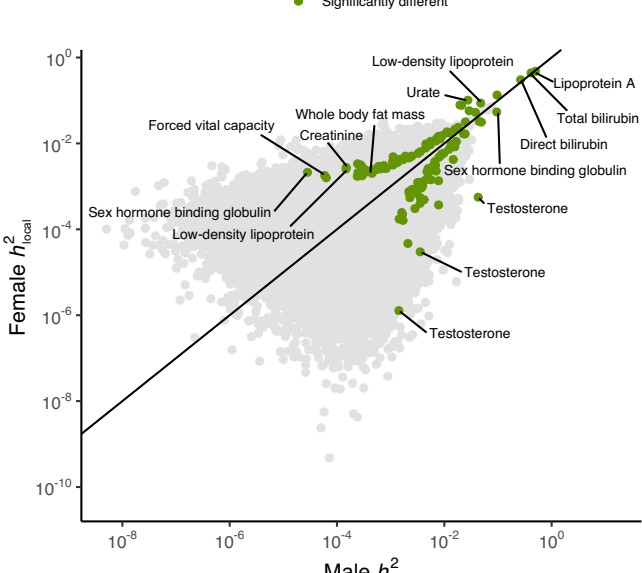

**Fig. 1 | Local heritabilities ($h^2$) estimated with LAVA for 2495 loci across 157 traits in males and females.** To test for significantly different heritability estimates, a Bonferroni-corrected threshold of $p = 0.05/(2495 \times 157) = 1.28e\text{-}07$ was used.

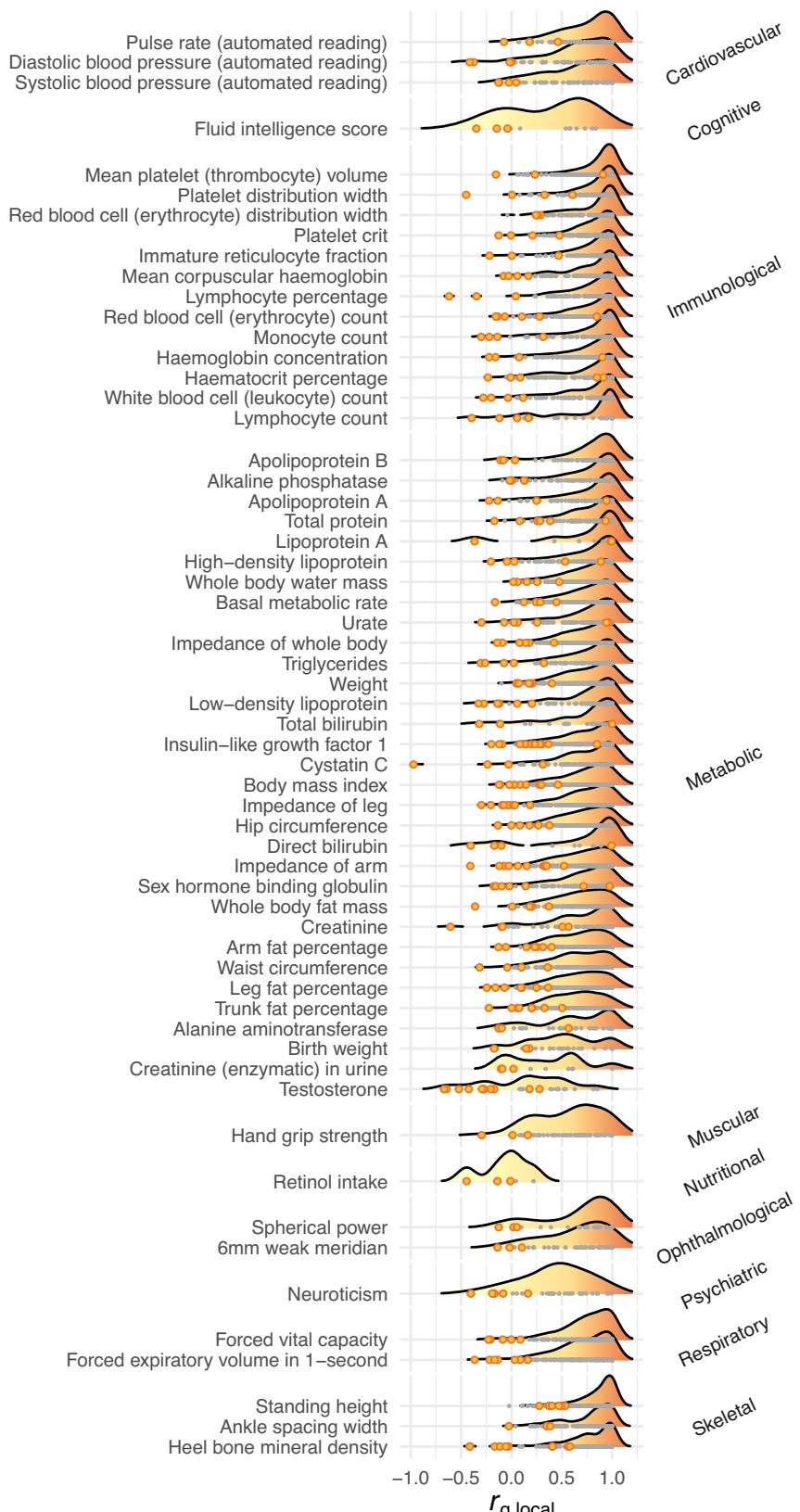

**Fig. 2 | Density plots of local genetic correlations ($r_{g,local}$s) estimated with LAVA.** Depicted are all traits with at least three loci that are significantly different from one at a Bonferroni-corrected significance threshold of $p < 0.05/11259 = 4.44e-06$ (this minimum number of significant loci was chosen to aid visualization; see Supplementary Fig. 4 for the number of significant loci for all traits). Grey dots are loci not significantly different from one and orange dots are loci significantly different from one.

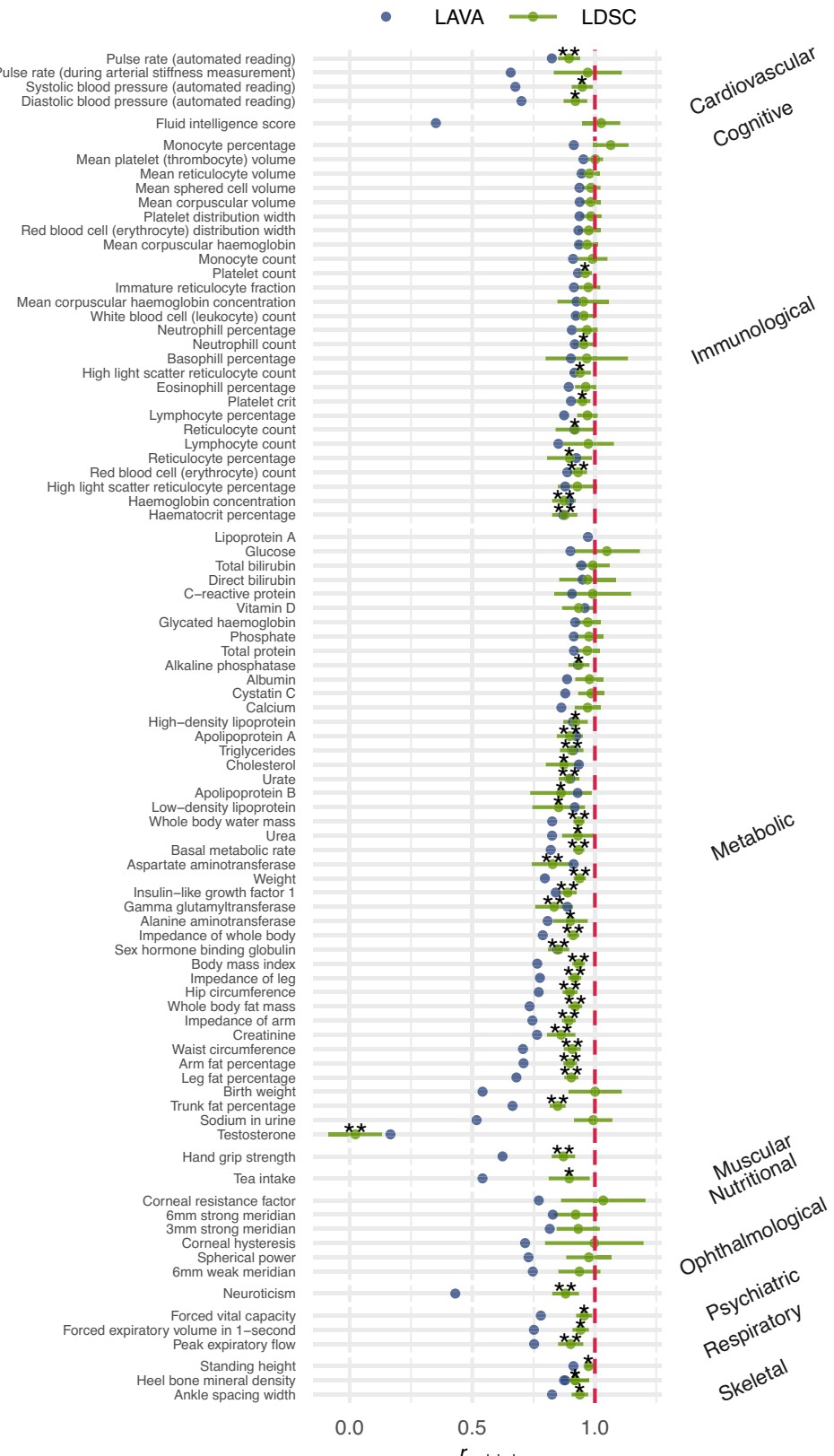

**Fig. 3 | Global genetic correlations ($r_{g, global}$) estimated with LD Score Regression and the variance-weighted mean of local genetic correlations estimated for LAVA.** The error bars for LDSC estimates depict standard errors. The depicted LAVA weighted means are not true estimates of $r_{g, global}$s and therefore have no associated $p$-values or standard errors. * nominal significance at $p < 0.05$ ** Bonferroni-corrected significance at $0.05/149 = 3.34e-04$.

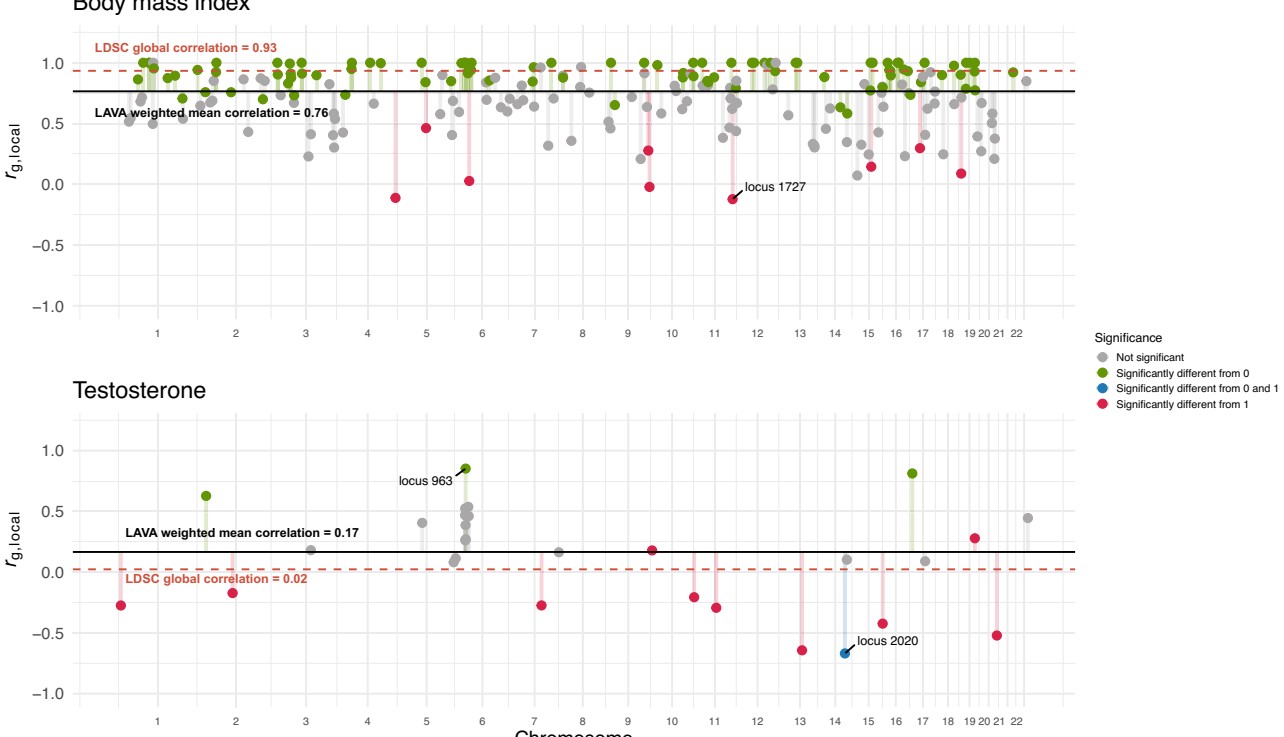

**Fig. 4 | LAVA local genetic correlation ($r_{g, local}$) results for BMI and testosterone.** While the LDSC-$r_{g, global}$ estimate for BMI is close to one, many loci have $r_{g, local}$s that strongly and significantly diverge from one (e.g., locus 1727). In contrast, the testosterone LDSC-$r_{g, global}$ estimate is close to zero, while some loci are strongly negative and others positive (e.g., locus 963 and 2020). Similar plots for all other traits can be found in our Zenodo repository (https://doi.org/10.5281/zenodo.15213372). A Bonferroni-corrected significance threshold of $p < 0.05/11259 = 4.44e\text{-}06$ was used for $r_{g, local}$s.

absolute difference = 0.12). Out of 119 traits with LDSC-$r_{g, global}$ not significantly different from one, 84 had at least one $r_{g, local}$ that was significantly different from 1. Body mass index (BMI) has an LDSC-$r_{g, global}$ of 0.93, but locus 1727 (11:122589225:123398882) has a $r_{g, local}$ of −0.12 ($p = 4.03e\text{-}06$) (see Fig. 4). Reversely, testosterone has an LDSC-$r_{g, global}$ of zero, but has several loci that are strongly positive and negative. For example, locus 2020 (14:93386329:94892240) has a $r_{g, local}$ of −0.67 and is significantly different from one ($p < 1.00e\text{-}300$), while locus 963 (6:32454578:32539567) has a $r_{g, local}$ of 0.85 which is significantly different from zero ($p = 2.37e\text{-}08$) but not from one ($p = 0.17$).

## Equality of genetic effects

A recent study suggested that differences in the absolute magnitude of genetic effects could strongly contribute to phenotypic sex differences[18]. As such, we extended LAVA to test for equality of genetic effects (see "Methods" for details). This is a more specific test for differences in genetic architecture because it not only tests for equality of the direction of genetic effects and their relative magnitudes (i.e., correlations) but also for equality of their absolute magnitude. For example, a locus with SNP effects with the same direction and pattern of effects between the sexes but where the effects are twice as large in males would have a $r_{g, local}$ of one, even though the SNP effects all differ. Equality of genetic effects implies that the $r_{g, local}$ is one and that the SNP effects are the same. As such, a test of equality of genetic effects is more specific than testing $r_{g, local}$ equal to one. The equality test is performed on two effect-size scales, the raw scale of the phenotype (equality$_{raw}$) and a standardized scale (equality$_{std}$), where the effect sizes are scaled by the standard deviation of the phenotype. When the genetic correlation in a tested locus is not significantly different from one, it is possible to make inferences about the $h^2_{local}$ and

phenotypic variance differences between the sexes based on the results of the two scales of the equality test (see Fig. 5). This is not the case when $r_{g, local}$ is significantly different from one because, in that instance, the equality test will tend to be significant even if no difference in $h^2_{local}$ or phenotypic variance exists (i.e., the individual SNP effects differ, even though the total genetic variance is the same). Moreover, as seen in the two middle columns, the two scales can only diverge if the phenotypic variances differ between the sexes.

Out of 157 traits, 151 and 152 had at least one locus that significantly ($p < 0.05/(2495 \times 157) = 1.28e\text{-}07$) differed between males and females in the equality$_{raw}$ and equality$_{std}$ test, respectively. Out of these, 45 loci across 23 phenotypes had $r_{g, local}$ not significantly different from one. The median number of significant loci per trait was 3.5. Not all significant loci in the equality tests overlap with GWAS hits. Out of 606 significant equality$_{raw}$ loci, 89 contained genome-wide significant SNPs in females and 130 in males. Out of 512 significant equality$_{std}$ loci, 64 contained genome-wide significant SNPs in females and 69 in males.

Because the equality test can be applied to arbitrarily small loci, we repeated the analysis using gene boundaries as locus definitions to improve their biological interpretation. However, we note that applying it to smaller loci while increasing the Bonferroni correction using the number of genes (i.e., $0.05/(18576 \times 157) = 1.7e\text{-}08$) comes at the expense of power. Of all 157 traits, 55 had at least one gene that significantly differed between males and females on at least one scale. Of these 55 traits, the median number of significant genes was 2. The correlation of $p$-values between equality$_{raw}$ and equality$_{std}$ across all traits and genes was 0.98, likely because most genes did not differ between the sexes on either scale and because the phenotypic variances were mostly similar (mean female-to-male ratio: $0.95 \pm SE\ 0.03$), except for testosterone (female-to-male ratio: 0.03) and oestradiol

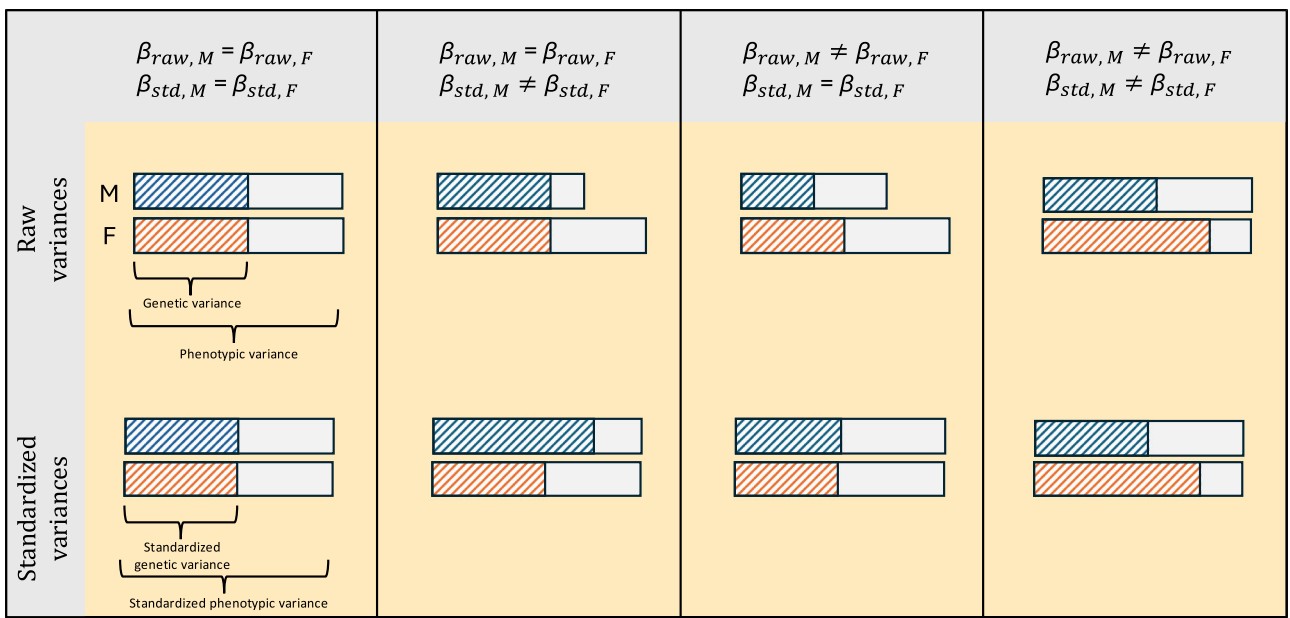

**Fig. 5 | Illustration of the scales of genetic effects when $r_{g, local} = 1$.** When genetic effects are the same on both the raw and standardized scale, the phenotypic and genetic variance will be equal between males (M) and females (F) (first column). When genetic effects differ on the standardized scale, but not the raw scale, phenotypic variances and heritabilities differ (second column). When genetic effects differ on the raw, but not the standardized scale, phenotypic variances differ while heritabilities are the same (third column). When both the raw and standardized genetic effects differ, heritabilities differ while phenotypic variances may or may not differ (fourth column). We note that when $r_{g, local} \neq 1$, genetic effects may differ under any of these scenarios.

(female-to-male ratio: 36.42). However, we identified 85 genes across 10 traits with divergent results in the equality$_{raw}$ and equality$_{std}$ test (i.e., one test was significant at $p < 1.7e\text{-}08$, while the other had $p > 0.01$). For sex hormone binding globulin (SHBG), three neighboring genes (i.e., NRBF2, JMJD1C, and REEP3) showed very similar and strong sex differences with equality$_{std}$ ($p < 1.3e\text{-}34$), but not with equality$_{raw}$ ($p \geq 0.7$). The phenotypic variance of SHBG was 3.6 times larger in females. Therefore, the heritability was expected to be larger in males (see column 2 in Fig. 5). Indeed, the heritability was significantly different ($p < 1e\text{-}08$) and more than 3 times larger in males for all three genes ($h^2_{local, females} = 0.3\%$, $h^2_{local, males} = 1.1\%$). JMJD1C has been hypothesized to impact SHBG levels via thyroid hormones that affect HNF4A[22], and while this seems to apply to both males and females, our results suggest this to be a much stronger effect in males. Reversely, for high-density lipoprotein (HDL), the gene CETP showed strong sex differences with equality$_{raw}$ ($p = 3.62e\text{-}32$) but not with equality$_{std}$ ($p = 0.02$). As expected (see column 3 in Fig. 5), the heritabilities did not significantly differ ($p = 0.07$, $h^2_{females} = 4.6\%$, $h^2_{males} = 4.3\%$). The phenotypic variance in females was 1.5 times larger than in males, which explains the observed difference with equality$_{raw}$.

Next, we tested whether significant loci in the equality test can highlight biological differences between the sexes for testosterone, diastolic blood pressure, and low-density lipoprotein (LDL). We also applied FLAMES to predict the most likely causal gene for each risk locus in males and females for these three traits. As a positive control, we highlighted results for testosterone because its biology is well-understood and known to be markedly different between males and females[13,16]. Whether or not a gene has a known function in testosterone biology strongly predicted association strength in the equality test ($p = 8.13e\text{-}09$). Reversely, nearly every significant locus in the gene-based equality test highlighted known biological differences in testosterone biology (see Fig. 6). Most testosterone GWAS hits did not overlap between males and females[16]: Out of 47 loci in the female GWAS and 112 loci in the male GWAS, four loci overlapped. Differences may also exist for overlapping loci. For one of the overlapping loci, we

found AKR1C3, a gene associated with testosterone synthesis and metabolism, to map to a GWAS hit in males (10:5012267:5512267) and females (10:4812752:5312752). However, the lead SNPs differ and are on opposite sides of AKR1C3, and both equality$_{raw}$ ($p = 5.19e\text{-}12$) and equality$_{std}$ ($p = 1.52e\text{-}14$) showed significant differences for this gene. There was no difference in $h^2_{local}$ ($p = 0.24$), but the 1 Mb locus that contained AKR1C3 had a $r_{g, local}$ of 0.18, which was significantly different from 1 ($p < 1e\text{-}8$) but not from 0 ($p = 0.17$). As such, while AKR1C3 contains signals in both males and females, the pattern of SNP associations is markedly different. Moreover, the gene prioritization tool FLAMES[23] predicted different causal genes at this locus, namely AKR1C2 for females and AKR1C4 for males, which we also found to be significantly different in the equality test (see Fig. 6). FLAMES predicted the same causal gene for the other three overlapping loci.

The major female sex hormone, oestradiol, has a very different genetic architecture from the major male hormone, testosterone. The sex-stratified GWASs of oestradiol contain little genetic signal, each identifying only one genome-wide significant risk locus and with $h^2_{global}$ estimates of ~2%. Moreover, the $r_{g, global}$ was not significantly different from one.

For diastolic blood pressure, we replicated a recent study that found the COL4A1/COL4A2 locus to be sexually dimorphic[24] (see Fig. 6). This locus is significant in the male GWAS but not the female GWAS. FLAMES[23] predicted COL4A1 to be the most likely causal gene for males. The equality test flags this gene to differ the most on both scales. However, we do not find evidence for differential effects of three other loci (i.e., PECAM1, NT5C2, MSTN) identified in the same study[24]. These loci did not reach statistical significance in an additional sex-by-genotype interaction analysis in the same study[24] and may thus be false positives. Furthermore, out of 101 female GWAS loci and 45 male GWAS loci, 26 loci overlapped. Of these 26 overlapping loci, 22 were predicted to have the same causal gene, while 4 were predicted to have different causal genes between males and females. However, none of these discordant genes were significant in the equality test.

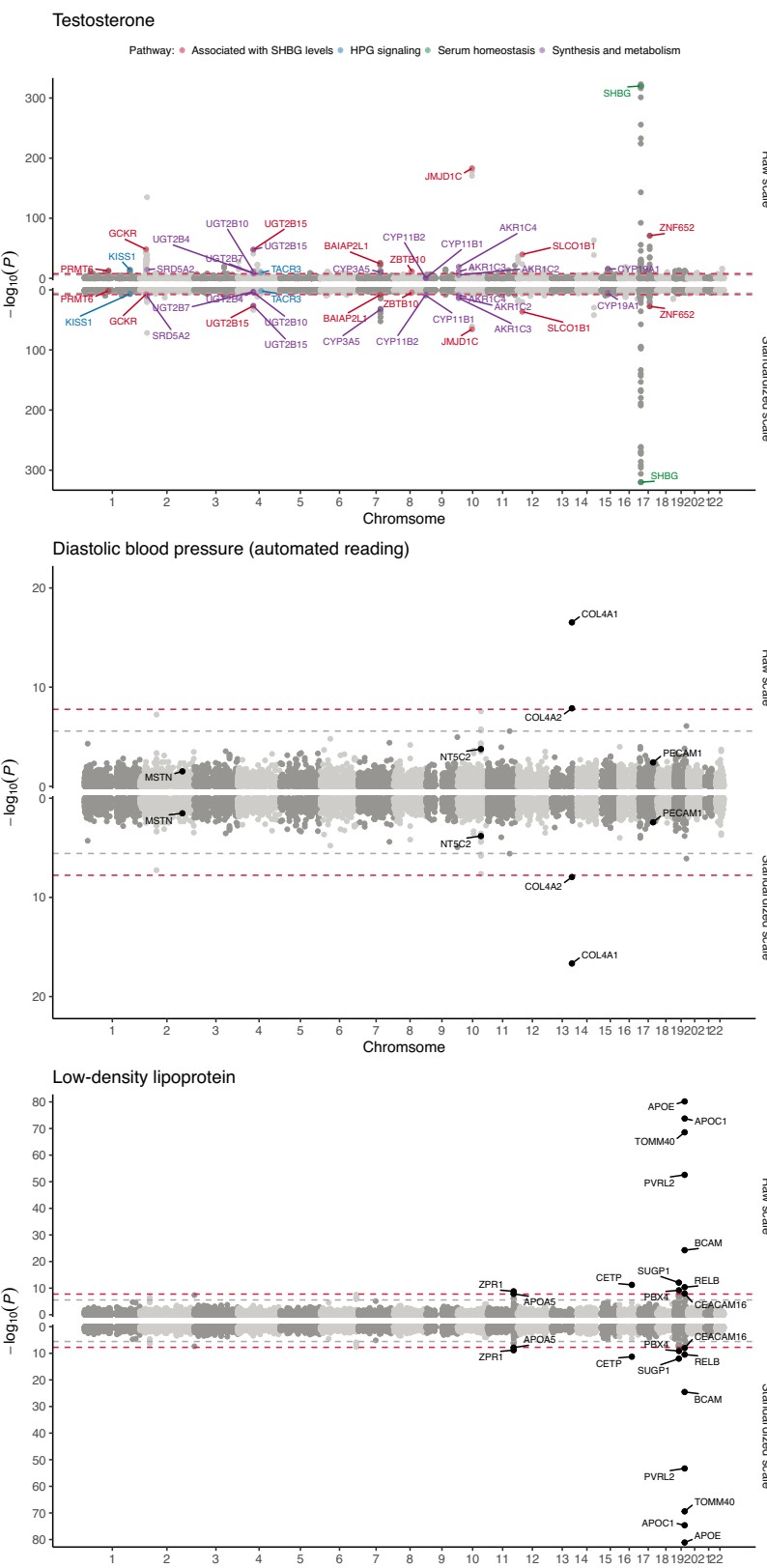

**Fig. 6 | Miami plots for the test of equal genetic effects for three traits.** The top part of the plots depict results for the raw scale, while the bottom part shows results for the standardized scale. The x-axis depicts chromosomal base-pair positions and the y-axis -log$_{10}$(p-values). For testosterone, all genes that are significant after Bonferroni correction ($p < 0.05/(2495 \times 157) = 1.28e{-}07$) and have a known function in testosterone biology are highlighted. For diastolic blood pressure, all genes that have been found to be sexually dimorphic in a recent study are highlighted. For low-density lipoprotein, all genes that are significant after Bonferroni correction are highlighted.

Lastly, APOE, which FLAMES predicted to be the causal gene at a GWAS hit for LDL in both males and females, shows a pronounced sex difference with both equality$_{raw}$ ($p = 6.78e\text{-}81$) and equality$_{std}$ ($p = 7.60e\text{-}82$) (see Fig. 6). While the $r_{g, local}$ of the locus (19:45040933:45893307) is 0.996, the $h^2_{local}$ of APOE, while large in both, is twice larger in females ($h^2_{local} = 6\%$) than in males ($h^2_{local} = 3\%$). Similarly, we found APOE to be sexually dimorphic for HDL, apolipoprotein B, and total cholesterol with the equality test on both scales, and triglycerides on the raw scale, and the $h^2_{local}$ are considerably larger in females (see Supplementary Fig. 7).

Lastly, out of 85 female GWAS loci and 56 male GWAS loci for LDL, we found 32 that overlapped between both. FLAMES predicted the same causal gene for 25 of these, and different causal genes for the other seven. However, only SUGP1 significantly differed between males and females in equality$_{raw}$ ($p = 8.22e\text{-}13$) and equality$_{std}$ ($p = 9.62e\text{-}13$). Interestingly, SUGP1 has previously been shown to be associated with coronary artery disease and cholesterol metabolism[25].

## Discussion

Early twin-based studies of genetic differences in heritability found few traits to be sexually dimorphic[26]. However, as GWAS sample sizes increase and some traits become saturated for genetic associations[27], an opportunity arises for sex-stratified analyses of common variants. Consequently, a recent large-scale analysis of hundreds of traits has found widespread, if small, differences across the genome[4]. Some traits, however, are particularly dimorphic, such as waist-to-hip ratio[28] and testosterone[16].

Many studies of genetic sex differences focus on $r_{g, global}$ and $h^2_{global}$ estimates[4,10,11] or manually compare the presence and absence of significant loci[15,16]. Such approaches may work well for traits with genome-wide sex differences but may miss traits with localized ones. Indeed, we found that 70% of traits for which LDSC estimates a $r_{g, global}$ close to one, have at least one locus with a $r_{g, local}$ different from one. Moreover, many of these had loci with negative $r_{g, local}$, which is rarely observed on a global level. This shows that $r_{g, global}$s between the sexes are not evenly distributed across the genome, but that loci of varying $r_{g, local}$s, some in opposite directions, combine to produce genome-wide correlations. This is particularly evident for testosterone, which has a $r_{g, global}$ of zero but several negative and positive $r_{g, local}$s.

A recent study has found that most observed genetic sex differences are not due to the direction and pattern of genetic effects (as measured with $r_g$) but due to the magnitude of genetic effects[18]. We built on this work by developing the equality of genetic effects test. Using this test, we provided insights chiefly in two ways. First, we tested for differences in the magnitude of genetic effects on the raw phenotypic, as well as the standardized scale. Comparing the scales can be informative for localized differences in genetic and phenotypic variances. We showed that differences in raw genetic effects can simply scale with differences in phenotypic variances, in which case heritabilities are expected to be the same between males and females. If the relative contribution of genetic effects on a phenotype is of interest, the standardized effects need to be considered. Secondly, we applied it to clearly defined genomic loci, and could thus test for differences in the magnitude of genetic effects within genes in addition to 1 Mb loci. Almost every trait we studied had at least one 1 Mb locus that differed between males and females. On a gene level, approximately a third of traits showed sex differences in at least one gene.

Combining LAVA bivariate $r_{g, local}$ and univariate $h^2_{local}$ analyses with the test of equal heritabilities, perfect $r_{g, local}$ and equality of genetic effects allows for very detailed descriptions of local genetic sex differences. In this way, we found APOE to have different magnitudes of genetic effects on the raw and standardized scales for several lipid-related phenotypes, while the $r_{g, local}$ was not different from one,

leading to $h^2_{local}$ estimates that are up to twice as large in females. As such, APOE must interact with male or female hormones or with external environmental variables (e.g., diet) to result in these divergent genetic effects. Future studies could compare whether these differences mediate sex-dimorphic effects of APOE on Alzheimer's Disease[29–33] and Cardiovascular Disease[34,35] risk. Moreover, we show for the AKR1C testosterone GWAS hit, how a $r_{g, local}$ of zero and significant equality tests can point to two different causal genes for males and females.

While for most overlapping GWAS loci of LDL, testosterone, and diastolic blood pressure between males and females, the same causal gene was predicted with FLAMES, some loci had divergent gene predictions. On a variant level, it is expected that the identity of causal variants does not differ between the sexes and that genetic sex differences are due to causal variant effect sizes only. On a gene level, we would therefore expect the same, namely that the same genes are causal and that merely the $h^2_{local}$ per gene may differ. However, several scenarios can give rise to diverging causal gene predictions. First, if GxS interactions are sufficiently strong, the effect of genes may start to appear qualitatively different. This seems to be the case for testosterone, which appears to be two separate phenotypes altogether in males and females with few overlapping risk loci, global genetic correlations of zero, and many genes significantly different in the equality test. Second, different causal genes may be predicted when a locus contains more than one causal gene, but the relative importance differs by sex, such that the top-ranking genes are not the same. Third, a locus may only reach genome-wide significance in one sex and may thus not yield a predicted causal gene in our analysis with FLAMES for the other sex (such as COL4A1 for diastolic blood pressure). Lastly, FLAMES selects the top-ranking gene as the most likely causal one, while the second- and third-ranking genes may be only slightly less likely. This may explain why most of these genes were not significantly different in the equality test. Moreover, FLAMES is expected to have 75% precision[36] and, as such, some uncertainty in the predictions is expected.

While nearly every trait showed some evidence for localized genetic sex differences, most loci for most traits did not differ. As such, while not tested in this study, we expect that genome-wide and additive polygenic predictions based on sex-stratified GWASs will not outperform those based on sex-combined GWASs. This is because sample sizes would be halved, thus drastically reducing power. However, other approaches that directly model GxS and polygenic covariance structures between males and females can improve risk prediction, particularly for traits with genome-wide sex differences, such as testosterone and waist-to-hip ratio[18].

There are several limitations to this study. First, LAVA computes local genetic correlations for relatively large genomic loci, which may contain multiple genes. Whether the genetic correlation is consistent across all or some genes is unknown. This is because genetic correlations cannot be reliably computed for small loci with insufficient genetic variance. However, compared to global genome-wide genetic correlations computed with methods such as LDSC[21], the 1 Mb loci in LAVA provide much more granularity. Second, we have only considered quantitative traits as work validating the newly developed tests for binary traits is still ongoing. Third, we only considered data from the UK Biobank. This is because it is the only large biobank we know for which sex-stratified GWAS summary statistics for quantitative traits have been computed and made publicly available for hundreds of traits. We urge the authors of future GWASs to release sex-stratified summary statistics. Lastly, loci that are significantly different between males and females do not necessarily contain significant GWAS hits. As such, these loci may contribute only minimally to phenotypic differences. Heritability estimates of the given locus can be evaluated to mitigate this, or loci may be filtered to those containing genome-wide significant SNPs.

## Methods

### Quality control of summary statistics

Summary statistics for 157 traits (see Supplementary Data 1) were downloaded from the Nealelab (https://github.com/Nealelab/UK_Biobank_GWAS). Only summary statistics for untransformed (i.e., raw) and quantitative phenotypes were downloaded. Phenotypes that were deemed too similar were filtered out, such that only one phenotype remained (e.g., hand grip strength left vs. right; see Supplementary Data 1 for a full list of traits that were considered and those that have been selected). Additionally, phenotypes that were suspected of having a minimal genetic basis were removed (e.g., "age at recruitment"). Details on how the GWAS was conducted and which quality control filters were applied can be found on the Nealelab website (https://github.com/Nealelab/UK_Biobank_GWAS). Briefly, sex-combined and sex-stratified GWASs were performed in 337,199 individuals of British ancestry using a linear regression model in Hail (https://github.com/hail-is/hail). For the sex-stratified analyses, the first 20 principal components, age, and $age^2$ were included as covariates. For initial quality control, all SNPs with imputation quality score below 0.8, minor allele frequency (MAF) below 0.1% (except for VEP[37] annotated SNPs), and Hardy–Weinberg-Equilibrium $p$-values below 1.00e-10 (except for VEP[37] annotated SNPs with MAFs below 0.1%) were removed. Considering that this database includes summary statistics that were generated on a large-scale without trait-specific quality control procedures, we additionally applied a strict MAF filter of 10% to ensure only well-imputed and reliable SNPs went into the analysis. This resulted in approximately five million SNPs for each trait. Manhattan and qq-plots, as well as LDSC Regression intercepts and Genomic Control statistics, were additionally inspected to ensure high-quality summary statistics.

### Genome partitioning

The genome was partitioned into blocks in the same way as in ref. 20, the method for which is described there in full. In brief, this partitioning method aims to divide the genome into smaller blocks of roughly equal size while minimizing the level of LD between them. It does so by recursively splitting the genome (starting at whole chromosomes), each time selecting the breakpoint for which the local LD between SNPs across that breakpoint is lowest. This process is repeated until no further valid breakpoints can be found; a breakpoint is invalid if it results in blocks containing fewer than the minimum number of SNPs specified (in the reference data) or if the level of LD across the breakpoint is too high.

The European panel of 1000 Genomes (phase 3)[38] was used to compute this partitioning, filtering out SNPs with a MAF lower than 1%. The minimum number of SNPs per block was set to 2500, resulting in 2495 blocks of, on average, about 1 Mb in size. Note that although the method aims to minimize the LD between blocks, some level of LD will generally still exist between adjacent blocks, and as such, they cannot be considered fully independent of each other.

### Local heritability, genetic correlation, and equality of genetic effects analysis

LAVA v0.0.7 scripts (https://github.com/josefin-werme/LAVA) were used to process all loci and compute univariate $h^2_{local}$ and bivariate $r_{g, local}$ estimates. All LAVA analyses were performed on 2495 semi-linkage disequilibrium independent blocks of approximately 1 Mb in size (https://github.com/josefin-werme/LAVA). Local genetic correlations were only estimated for loci with sufficient evidence of heritability in both males and females. This is because a genetic correlation cannot exist if no genetic variance is present, and filtering loci with little to no heritability improves computational efficiency while reducing the multiple testing burden. To this end, we applied a $h^2_{local}$ threshold of $p < 1.00e-04$. Using this threshold reduced the number of loci by 93% while retaining 82% of loci with genome-wide significant SNPs in both

males and females (see Supplementary Data 3). Because the summary statistics were based on individuals of British ancestry, we used the European sample of phase 3 of 1000 Genomes[38] as the Linkage Disequilibrium reference sample (https://ctg.cncr.nl/software/lava). All genomic coordinates refer to human genome build 37. Finally, it was tested if the $h^2_{local}$s differed, if $r_{g, local}$s significantly differed from one, and if the genetic effects significantly differed between the sexes.

### LAVA model

A brief overview of the LAVA model is given here, a full description can be found in Werme et al.[20]. LAVA assumes a linear model of the form $Y_p = X\alpha_p + \varepsilon_p = W\delta_p + \varepsilon_p$ for each continuous phenotype $p$, with standardized genotype matrix $X$ of SNPs in the locus being analysed, standardized phenotype vector $Y_p$, and residual variance $\eta^2_p = \text{var}(\varepsilon_p)$. The predictor matrix $W$ contains standardized principal components obtained from $X$, and these are used to deal with the high degree of collinearity in $X$. In practice, estimates for $\hat{\delta}_p$ and $\eta^2_p$ are obtained by reconstructing the linear regression model from GWAS summary statistics and the LD structure of $X$ from genotype reference data. Under the model, the estimates $\hat{\delta}_p$ are distributed as $\hat{\delta}_p \sim \text{MVN}(\delta_p, \sigma^2_p I_K)$, with sampling variance $\sigma^2_p = \frac{\eta^2_p}{N_p - 1}$, sample size $N_p$, and with $K$ the number of principal components.

The local genetic component $G_p = X\alpha_p = W\delta_p$ is defined for each phenotype, which are combined into a matrix $G$ of local genetic components for all phenotypes in the analysis. In the context of this study, the same phenotype for each sex is treated as two separate phenotypes. The quantity of interest is the local genetic covariance matrix $\Omega = \text{cov}(G)$, from which the local genetic components can be computed as $\rho_{pq} = \frac{\omega_{pq}}{\sqrt{\omega^2_p \omega^2_q}}$, for each pair of phenotypes $p$ and $q$. Since the phenotype vectors are assumed to be standardized, the local heritability for each phenotype $p$, which is the explained variance of the linear regression model, is equal to the variance of $G_p$, which are the diagonal elements of $\Omega$. A method of moments estimator is used to obtain an estimate $\hat{\Omega}$.

### Testing equality of genetic effects

Strict homogeneity of the local genetic structure of the phenotype across sexes can be tested using a null hypothesis $H_0 : \delta_M = \delta_F$, with $\delta_M$ and $\delta_F$ the genetic effect vectors for men and women for that phenotype, respectively (this is also equivalent to testing $H_0 : \alpha_M = \alpha_F$). Under this null model, the difference $\hat{D} = \hat{\delta}_M - \hat{\delta}_F$ is distributed $\hat{D} \sim \text{MVN}(0, (\sigma^2_M + \sigma^2_F)I_K)$, and it, therefore, follows that the test statistic $\frac{\hat{D}^T \hat{D}}{\sigma^2_M + \sigma^2_F}$ has a $\chi^2_K$ distribution, which can be used to obtain a p-value. Because these genetic effect vectors are defined on a standardized scale, the null hypothesis $H_0 : \delta_M = \delta_F$ implies that the correlations of all of the SNPs in $X$ with the phenotype (as well as the local heritabilities generally) are the same for each gender. However, homogeneity of genetic effects can also be defined on the natural scale of the phenotype instead, which implies that a change in genotypes results in the same amount of change (on the natural scale) in the phenotype. Writing $Y^*_p = Y_p S_p$, with $Y^*_p$ the phenotype on its natural scale and $S_p$ its standard deviation, the corresponding genetic effects on this scale would be $\delta^*_p = \delta_p S_p$, and equality on this scale can be tested using the null hypothesis $H_0 : \delta^*_M = \delta^*_F$, which is equivalent to testing $H_0 : S_M \delta_M = S_F \delta_F$. The corresponding difference vector $\hat{D}^* = S_M \hat{\delta}_M - S_F \hat{\delta}_F$ is distributed $\hat{D}^* \sim \text{MVN}(0, (S^2_M \sigma^2_M + S^2_F \sigma^2_F)I_K)$ under this null, and the test statistic $\frac{\hat{D}^{*T} \hat{D}^*}{S^2_M \sigma^2_M + S^2_F \sigma^2_F}$ again has a $\chi^2_K$ distribution. We

note that, at present, this test has only been evaluated for quantitative phenotypes. Simulations showed that type-1 error rates were well controlled (see Supplementary Data 2 and "Methods: Type-1 error simulations").

## Testing equality of local heritability

As the local heritability $h_p^2$ of a phenotype $p$ equals the variance of $G_p$, it can be expressed as $h_p^2 = \text{var}(G_p) = \frac{\delta_p^T W^T W \delta_p}{N_p - 1} = \delta_p^T \delta_p$. Moreover, $\eta_p^2 = 1 - h_p^2$. Since $\hat{\delta}_p \sim \text{MVN}(\delta_p, \sigma_p^2 I_K)$, the statistic $T_p = \frac{\hat{\delta}_p^T \hat{\delta}_p}{\sigma_p^2} = \frac{(N_p - 1)}{1 - h_p^2} \hat{\delta}_p^T \hat{\delta}_p$ has a noncentral $\chi_K^2$ distribution with non-centrality parameter $\lambda_p = \frac{\delta_p^T \delta_p}{\sigma_p^2} = (N_p - 1)\frac{h_p^2}{\eta_p^2} = (N_p - 1)\frac{h_p^2}{1 - h_p^2}$. Defining $C_p = \frac{h_p^2}{1 - h_p^2}$, the expected value of $T_p$ can be expressed as $E[T_p] = K + \lambda_p = K + (N_p - 1)C_p$.

Under the null hypothesis $H_0 : h_M^2 = h_F^2 = h^2$ for a shared $h^2$, the expected value of the difference in test statistics is $E[T_M - T_F] = E[T_M] - E[T_F] = K + (N_M - 1)C - (K + (N_F - 1)C) = C(N_M - N_F)$, with $C = \frac{h^2}{1 - h^2}$. To test this null hypothesis, we first estimate the shared $h^2$ parameter as the sample size weighted mean of the sex-specific estimates, i.e., $\hat{h}^2 = \frac{1}{N_M + N_F}(N_M \hat{h}_M^2 + N_F \hat{h}_F^2)$. We then define the test statistic $T_D^{(\text{obs})} = T_M^{(\text{obs})} - T_F^{(\text{obs})} = \frac{1}{1 - \hat{h}^2}((N_M - 1)\hat{\delta}_M^T \hat{\delta}_M - (N_F - 1)\hat{\delta}_F^T \hat{\delta}_F)$, and generate draws of $T_D^{(\text{draw})}$ by separately sampling $T_M^{(\text{draw})}$ and $T_F^{(\text{draw})}$ from $\chi_K^2$ distributions with noncentrality parameters of $(N_M - 1)\hat{C}$ and $(N_F - 1)\hat{C}$ with $\hat{C} = \frac{\hat{h}^2}{1 - \hat{h}^2}$, and taking their difference. An empirical p-value is then computed as $2 \times \Pr(T_D^{(\text{draw})} < T_D^{(\text{obs})})$ if $T_D^{(\text{obs})} < \hat{C}(N_M - N_F)$, and $2 \times \Pr(T_D^{(\text{draw})} > T_D^{(\text{obs})})$ otherwise. An adaptive sampling procedure was used for this, starting at an initial 10,000 draws of $T_D^{(\text{draw})}$, and increasing these up to a maximum of 100 million draws for lower p-values.

We note that, at present, this test has only been evaluated for quantitative phenotypes. Simulations showed that type-1 error rates were well controlled (see Supplementary Data 2 and "Methods: Type-1 error simulations"). Further simulations also indicate that if sample sizes differ, power to detect a difference in local heritability will be somewhat greater if the higher local heritability is in the smaller of the two samples, rather than in the larger sample (see Supplementary Fig. 3 and "Methods: Power simulations").

Note that the null hypothesis of equal local heritabilities, $H_0 : h_M^2 = h_F^2$, is also implied by the null hypothesis $H_0 : \delta_M = \delta_F$ of equality of standardized genetic effects, though the reverse is not true. By contrast, under the null hypothesis $H_0 : S_M \delta_M = S_F \delta_F$ of equality of natural scale genetic effects, heritabilities can only be equal if $S_M = S_F$.

## Testing perfect correlation of local genetic signal

To test the null hypothesis $H_0 : \rho_{pq} = \rho_0$ of a perfect local genetic correlation (where $\rho_0 = 1$) between two phenotypes, we generalized the base LAVA model (which tests $H_0 : \rho_{pq} = 0$ by default). To do so, we defined the test statistic $T_\rho = \hat{\omega}_{pq} - \sqrt{\hat{\omega}_p^2 \hat{\omega}_q^2} \rho_0$, i.e., the estimate of the local genetic covariance $\hat{\omega}_{pq}$ minus its estimated expected value given a specific null value $\rho_0$ for the local genetic correlation. This reverts to the $T_\rho = \hat{\omega}_{pq}$ used in the original LAVA implementation when $\rho_0 = 0$. The full matrix estimate $\hat{\Omega}$ has a noncentral Wishart distribution with $K$ degrees of freedom, scale parameter $\Sigma = \begin{pmatrix} \sigma_p^2 & 0 \\ 0 & \sigma_q^2 \end{pmatrix}$ and noncentrality parameter $\Lambda = \Sigma^{-0.5} \Omega \Sigma^{-0.5}$ (see also Werme et al.[20]). Filling in these

parameters with the null value $\rho_0$ and the sample estimates, draws of $T_\rho$ can therefore be generated by sampling values of $\hat{\Omega}$ and computing the corresponding values of $T_\rho$. An empirical p-value can then be computed as $\Pr\left(\left|T_\rho^{(\text{draw})}\right| > \left|T_\rho^{(\text{obs})}\right|\right)$. An adaptive sampling procedure was used for this, starting at an initial 1,000 draws of $T_D$, and increasing these up to a maximum of 100 million draws for lower p-values.

For this study, we specifically tested the null hypothesis $H_0 : \rho_{MF} = 1$, to determine whether the sex-stratified local genetic signals for a phenotype were perfectly correlated. This null hypothesis is true if $\delta_M = c\delta_F$ for any arbitrary value $c > 0$, and is therefore implied by the null hypothesis of equal genetic effects at both the standardized ($c = 1$) as well as natural ($c = \frac{S_F}{S_M}$) scale, which shows that the $\rho_{MF} = 1$ test can be seen as a generalization of the equality of genetic effect tests. There is no logical relationship between this null hypothesis and the null hypothesis of equal local heritability since a local genetic correlation of one does not require local heritabilities to be equal, and equality of local heritabilities can exist at any value of the local genetic correlation. We note that, at present, this test has only been evaluated for quantitative phenotypes. Simulations showed that type-1 error rates were slightly deflated at low sample sizes but well controlled at higher sample sizes and heritabilities (see Supplementary Data 2 and "Methods: Type-1 error simulations").

## Type-1 error simulations

Type 1 error simulations were performed by first generating 250 independent principal components $W$ for the desired sample size (with 250 being generally representative of the number of genetic principal components per block in the LAVA analyses). Continuous phenotypes were then simulated by first setting the true (raw scale) genetic effect vectors $\delta_p^*$ and $\delta_q^*$ for each of the two phenotypes $p$ and $q$ according to the desired null model, computing the genetic components $G_p = W\delta_p^*$ and $G_q = W\delta_q^*$, and adding normally distributed noise to these with variance set to obtain a specified phenotypic variance to obtain simulated phenotypes $Y_p$ and $Y_q$. These were each standardized and then regressed on $W$ to obtain the estimates $\hat{\delta}_p$ and $\hat{\delta}_q$ of the standardized effects $\delta_p = \frac{\delta_p^*}{\text{SD}(Y_p)}$ and $\delta_q = \frac{\delta_q^*}{\text{SD}(Y_q)}$, which were then used as input for the test being evaluated.

For each condition, 10,000 repeats were generated, and type 1 error rates were computed for $\alpha$ values of 0.05 and 0.001 as the proportion of repeats for which $p < \alpha$. For each of the four evaluated tests, for the first phenotype, the variance $\text{var}(Y_p)$ was always set to one, and the heritability $h_p^2$ was varied across 0%, 0.1%, 0.5%, and 1%. Sample sizes were set to either 10,000 or 50,000.

For the test of equality of genetic effects, simulations were performed for equality under the raw scale ($H_0 : \delta_p^* = \delta_q^*$) as well as the standardized scale ($H_0 : \delta_p = \delta_q$), with the variance of the second phenotype was set either to $\text{var}(Y_q) = \text{var}(Y_p) = 1$ or to $\text{var}(Y_q) = 2 \times \text{var}(Y_p)$. For the test of equality of local heritabilities, the null model was set to $H_0 : h_p^2 = h_q^2$, with $\delta_p \neq \delta_q$ and $\text{var}(Y_q) = \text{var}(Y_p) = 1$.

Finally, for the test of perfect local genetic correlations, the null model $H_0 : \text{cor}(G_p, G_q) = 1$ was used. This is equivalent to $H_0 : \delta_q = c\delta_p$ for an arbitrary positive value of $c$, which implies $h_q^2 = c^2 h_p^2$. The parameter $c$ was set to either 1 or $\sqrt{2}$, and $\text{var}(Y_q) = \text{var}(Y_p) = 1$. For these simulations, the $h_p^2 = 0$ conditions were omitted, as the local genetic correlation is not defined if no local genetic variance is present for either of the phenotypes. Exact type-1 error rates for each test and condition can be found in Supplementary Data 2.

## Power simulations

Additional simulations were performed for the test of equal heritabilities to evaluate possible asymmetry in power due to differences in sample sizes. For each simulation condition, a base sample size $N_p$ for the first sample was set (using values 10,000, 20,000, 100,000, and 200,000). The sample size for the second sample was then set to $N_q = RN_p$, setting the sample size ratio $R$ at either 1.25 or 2. As in the type 1 error rate simulations, the number of genetic principal components $K$ was set to 250. A non-zero heritability value $h^2$ was then specified for one of the two samples (using values 0.01% to 0.1% (increments of 0.01), 0.12%, 0.15%, 0.2%, 0.5%, and 1%), while setting the heritability in the other sample to zero.

To perform the actual simulations, values were generated by simulating a random variable $D_p \sim \chi^2_{K,\lambda_p}$ with noncentrality parameter $\lambda_p = (N_p - 1)\frac{h_p^2}{1-h_p^2}$, then setting $\hat{\delta}_p^T \hat{\delta}_p = D_p \sigma_p^2$ and $\hat{h}_p^2 = \hat{\delta}_p^T \hat{\delta}_p - K\sigma_p^2$ with sampling variance $\sigma_p^2 = \frac{1-h_p^2}{N_p-1}$. Values for $\hat{\delta}_q^T \hat{\delta}_q$ and $\hat{h}_q^2$ were generated in the same way. The equality test was then performed for these simulated values as specified above in "Methods: Testing equality of local heritability", using a fixed 10,000 draws to compute the p-value.

For each condition, 100,000 values were simulated with $h_p^2 = h^2$ (and $h_q^2 = 0$), and the power was computed using significance thresholds of 0.05 and 0.001. This was then repeated with $h_q^2 = h^2$ (and $h_p^2 = 0$). Results for these simulations are shown in Supplementary Fig. 3.

## Global genetic correlations

We used LDSC Regression v1.0.1[21] to compute $r_{g,\text{global}}$ between males and females for 157 traits to describe the average effect of pleiotropy across the whole genome. LD scores based on the European sample of 1000 Genomes were used. Out of 157 traits, 8 had insufficient $h^2$ estimates for correlations to be computed. As such, we used a Bonferroni-corrected significance threshold of $0.05 / 149 = 3.34\text{e-}04$. We computed the following t-statistic to test the null hypothesis of a perfect correlation[4]: $t = \frac{r_{g,global} - 1}{SE}$, where SE is the LDSC-estimated standard error of $r_{g,global}$. With LAVA, we computed the inverse-variance weighted mean of $r_{g,\text{local}}$s. Only loci that exceeded a $h^2_{\text{local}}$ p-value threshold of $1.00\text{e-}04$ were used to compute this mean.

## FLAMES

We used the SusieR[39] implementation in PolyFun[40] to fine-map FUMA-defined risk loci[41]. We allowed for a single causal variant per locus. Fine-mapping results were transformed to 95% credible sets by including the smallest number of variants whose posterior inclusion probability sum to at least 0.95. We generated MAGMA[42]-Z scores using a UK Biobank LD reference panel and the gene annotations used in the original PoPS publication[43]. PoPS scores were generated using the previously generated MAGMA Z-scores using PoPS v0.2. We created gene level annotations using FLAMES[23] annotate, with the generated MAGMA Z-scores, PoPS scores, and 95% credible sets as input. Gene prioritization was performed using FLAMES v1.0.0. We prioritized genes with a FLAMES score above the recommended threshold of 0.05. FLAMES prioritizes a single gene per locus.

## Reporting summary

Further information on research design is available in the Nature Portfolio Reporting Summary linked to this article.

## Data availability

Sex-stratified GWAS summary statistics: https://github.com/Nealelab/UK_Biobank_GWAS. LAVA locus definition file: https://github.com/josefin-werme/LAVA. 1000 Genomes LD reference file for LAVA: https://ctg.cncr.nl/software/lava. Scripts, plots, and results for all LAVA analyses and all 157 traits, and FLAMES results for Testosterone, Diastolic blood pressure, and LDL direct: https://doi.org/10.5281/zenodo.15213372.

## Code availability

Scripts for testing equality of genetic effects, equality of local heritability, and perfect correlation of local genetic signal can be downloaded from: https://doi.org/10.5281/zenodo.15213372.

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

## Acknowledgements

D.P. and M.S. are supported by the Netherlands Organization for Scientific Research - Gravitation project 'BRAINSCAPES: A Roadmap from Neurogenetics to Neurobiology' (024.004.012), and D.P., E.U., and C.d.L. are supported by the European Research Council advanced grant 'From GWAS to Function' (ERC-2018-ADG 834057).

## Author contributions

D.P. and E.U. conceived of the project. E.U. performed most analyses. C.d.L. wrote the R code for the equality of genetic effects, perfect local genetic correlation, and equality of local heritability tests, and performed simulations to probe their statistical properties. M.S. performed gene prioritization analyses with FLAMES. E.U. wrote the paper, and C.d.L. and M.S. contributed to the "Methods" section. All authors discussed the results and commented on the paper.

## Competing interests

C.d.L. was funded by Hoffman-La Roche until July 2022. The other authors declare no competing interests.
