## [Transparent Peer Review file · Nature Communications]

Local Genetic Sex Differences in Quantitative Traits

Corresponding Author: Mr Emil Uffelmann

Version 0:

Reviewer comments:

Reviewer #1

(Remarks to the Author)

In this manuscript, Uffelmann et al. reported local sex differences in the genetic effects of 157 quantitative traits in the UK Biobank. They investigated sex differences in local heritability (h^2), genetic correlation (r_g), as well as the absolute differential effective size of r_g . Traits that show greater differences are phenotypes that are known to be sex dimorphism. Although sex differences in genetic effects at a regional scale have been sparsely studied, some of the discoveries in the manuscript have been previously reported in the same population (Elena et al., Nature Genetics, 2021). In fact, the genome-wide genetic effect comparison in the Nat Gen paper provides more biological insights than this manuscript, as it identifies specific SNPs/genes rather than regions, which may encompass multiple genes. Another question regarding this manuscript is the interpretation and validation of their findings. Whether the $G \times \text{Sex}$ differences could be causal to the observed sex differences in the traits, and the biological mechanisms underlying such differences, have not been comprehensively illustrated. Only a few known examples have been provided. Also, the manuscript lacks discussion on the limitations of their study and the comparison of their results to other relevant publications.

Here are some detailed comments:

1. Lacking interpretation of those sex differential regional genetic effects. Sex differences in h^2 and r_g in a region indicate that the genetic contribution to the trait differs between sexes in the given region. However, since each region may contain many genes and regulatory elements, the observed sex differences in r_g and h^2 do not provide further insight into causal discovery or biological mechanisms. It is uncertain how to interpret these differences, such as identifying the putative causal gene for the trait.
2. Limited in novel discoveries. In the manuscript, the author provided several examples of literature-reported evidence to support their discovery. While this serves as proof of concept, there is a lack of additional functional annotation for other discoveries, except for testosterone, which has been extensively studied before (Nat Med 26, 252–258 (2020)). Additional analysis is needed to characterize novel discoveries and improve the validity and impact of this work.
3. The rationale for choosing the h^2 threshold. The $h^2 p < 1e-4$ was used as the region selection criterion for further analysis. It is uncertain whether there are any genome-wide significant loci in the region or if any genome-wide significant loci would be missed by using this cutoff. To better define the rationale for choosing this cutoff, it is necessary to test how many credible sites under different cutoffs in a region should be examined. In fact, given that quantitative traits were investigated here, it is arguable that a more stringent cutoff should be used to avoid false positive discoveries, as continuous traits have greater power and often yield stronger GWAS signals than binary traits.
4. What is the interpretation of a negative r_g ? Does this imply an opposite genetic effect of the region? Does this imply a potential differential environmental effect on the trait between sexes? Additionally, what does the statement in line 91, " r_g is significantly different from zero," mean?
5. In the testosterone example, genes in regions were selected based on existing knowledge. However, whether the genes actually lie within the causal locus is unknown. This highlights the main limitation that regional test is limited in identifying putative causal genes/loci.

Reviewer #2

(Remarks to the Author)

Uffelmann and colleagues present a study to investigate the local genetic differences between males and females in 157 quantitative traits using the publicly available GWAS summary statistics in UKBB. They compared local heritability

estimates, estimated local genetic correlation, and tested the equality of genetic effects between sexes. They found that most of the quantitative traits that were studied have at least one locus that shows significant difference in males and females. They highlighted several examples, including loci for testosterone, the locus containing the gene MCPT2 for Cystatin C and the locus containing APOE for LDL-direct. While the results are interesting, there are some major issues that I would like to bring to the authors' attention.

Several sections within the manuscript were not presented with clarity.

1. The authors analyzed 157 traits in the UKBB, it is unclear how those quantitative traits were selected. For example, in the methods (Line 177), it reads "Additionally, phenotypes were removed that were suspected to have minimal genetic basis (e.g. 'age at recruitment')." What metrics were used here? Were global heritability estimates used for filtering?
2. Similarly, it would be great to have clear cutoffs for excluding or including trait and loci in analyses. For example, in Line 44, it reads "After the exclusion of loci with very large h^2_{local} (i.e. lipoprotein A, direct bilirubin, and total bilirubin)". Please clarify which cutoff for heritability was used.
3. Line 84, it reads "For many traits with LDSC- r_g , global close to one, we estimate some r_g , locals that are closer to zero and in some instances negative.". Please clarify the specific number of traits before describing examples.
4. The authors mentioned the power difference in females and males due to the female sampling bias in the UK Biobank and GWAS in females tend to have larger sample sizes than in males (Line 51). Please provide more descriptions about the sample size differences including which traits have largest differences.
5. Line 62, for locus and trait to have sufficient genetic signals, besides p-values for h^2_{local} , the magnitude is also important.
6. Can authors use the equality test to disentangle the reasons for loci to have different h^2_{local} ? Do those loci have different effect sizes or allele frequencies between sexes? For example, the locus 2181 (17:7264459:8554763) for testosterone was found to be significant in the equality test. What are the allele frequencies in males and females for this locus?
7. To understand the biological differences between the sexes, MAGMA gene analysis was performed. The description is unclear why the gene analysis was used here. How the co-localization was performed to connect the genes identified by MAGMA that are significantly associated in males and females separately to the equality tests. The authors described the pathways, to which genes are related to. The results could be biased without pathway enrichment analyses.
8. The r_g , local and h^2_{local} were estimated in this study in the LD independent blocks across the genome. How many of the loci identified to have differences in females and males are GWAS loci?
9. The authors extended LAVA to test for equality of genetic effects. Please evaluate the performance of the test, especially for the type I error rates or FDR using simulations
10. Because LAVA can also be used for binary phenotypes, can authors discuss why only quantitative traits were studied in this paper? Are there limitations in the current analyses so that binary phenotypes are not included?

Minor:

- Please cite LAVA at its initial appearance in the manuscript (Line 89)
- In Line 45, is the Pearson's correlation estimate reported? Please clarify.

Reviewer #3

(Remarks to the Author)

The manuscript "Local Genetic Sex Differences in Quantitative Traits" addresses an important area of genetic research, namely sex differences in the heritability of quantitative traits, for the first time looking at local heritability. Given the increasing focus on precision medicine, together with sample sizes that enable sex-stratified analyses, this is a timely piece of work that attempts to fill a genuine gap and which I think will be of broad interest. The authors find that while most loci do not differ in terms of direction or magnitude of effect across the sexes, almost every trait that was studied (157 quantitative traits) did have at least one locus that did show a difference. Overall, the results are interesting but there are significant limitations to this study.

General comments

It is not clear to me why the authors have chosen to only study sex differences in quantitative traits. It is also unclear, why only sex stratified GWAS analyses using UK BioBank data have been used. This should be explained.

Furthermore, the manuscript is very short given the complexity of the topic with a particularly short introduction and discussion. This means that readers without a background in the field would find it hard to contextualise the work, or fully understand the results and methods. Therefore, I think it would be sensible for the authors to extend the text, particularly the introduction and discussion.

Finally, only limited effort is made to dissect or understand the molecular basis of sex differences in local heritability and the implications for diagnostics or treatment e.g. could such findings point to the need for sex-stratified PRS or potentially explain sex differences in response to treatment. While I appreciate that these questions may be beyond the scope of the manuscript itself, they could be considered and discussed.

Specific comments

Focusing on the first analysis presented (Fig. 1)

- The autosomal genome is divided into 2495 'semi LD independent' loci of ~1Mb with h^2_{local} computed between all traits and all loci. While this method is referenced (Werne et al., 2022: 'An integrated framework for local genetic correlation

analysis') to understanding exactly how the partitioning was performed required reading supplementary methods of the Werne et al. paper. Given the importance of partitioning for local heritability analyses, it would be helpful for this information to be easier to find and the rationale explained.

Focusing on the second analysis (Fig. 2)

- In this analysis local genetic sex differences were interrogated using LAVA. However, it was not clear whether the phenotypes being studied were genetically correlated to each other and therefore whether sex differences were therefore expected to be found repetitively. I think it would be helpful to clarify this.
- Included in this analysis were all traits and locus pairs with $h^2_{\text{local}} P < 1e-04$. However, it is not clear why this P-value cut-off was used. There is no mention of this in the main text, nor in the methods where it is also mentioned (page 13). Could this be explained?
- Computed local genetic correlations (r_{glocal}) for males and females were generated and a test run for deviation from one, presumably perfect correlation. It would be helpful if this was explained and stated for the reader.

Focusing on the third analysis (Fig. 3 + Fig. 4)

- Next, to test that LAVA was producing valid results, computed global genetic correlations (r_{gglobal}) using LDSC were generated and compared. While finding good agreement between methods is helpful, given that the major innovation of this analysis is local heritability analyses the emphasis should be on traits with discrepancies. Consequently, Figure 4 is important in that it visualises the exact traits and genomic locations which have LAVA-LDSC discrepancies. For individual discrepancy examples mentioned in the text (page 6), it would be helpful to explicitly label them in figure 4.

Fourth analysis (Fig. 5)

- To further understand why r_{gglobal} and the weighted mean of r_{glocal} generated different magnitudes, they extended LAVA to test for the equality of genetic effects. This incorporates a test for equality of the direction of genetic effects and their relative magnitudes (i.e. correlations), but also for equality of their absolute magnitude. I thought this was a very interesting new metric. Is this functionality now formalised in the LAVA package and could that be stated?

Fifth analysis (Fig. 6)

- I thought use of testosterone as a positive control for the biological validation of the equality test was very valuable. In this context, it would also be helpful to highlight an example which is a positive control for females? Furthermore, power differences across the sexes are touched upon, but given the importance I wonder if this could be explored further and modelled.

Minor comments

Page 12: '(URLs)' in text rather than actual script repository link

Page 13: '(Data availability)' in text rather than link

Page 6: 'very close one' should be 'very close to one'

Version 1:

Reviewer comments:

Reviewer #1

(Remarks to the Author)

The revision includes additional analysis using the gene patronization approach FLAMES to further annotate regions of interest, enhancing the manuscript's rigor. However, the performance of this approach in identifying causal genes needs clarification. For instance, how many genes were identified per region? Do the number of genes and their effects differ between sexes? Quantitative illustrations would aid in interpreting this analysis.

To facilitate biological discovery, validating key findings with independent cohorts is crucial to extend the study's impact. Replicating top hits is necessary to confirm the existence of sex differential genetic effects. The lack of publicly available sex-stratified GWAS data limits replication. I suggest exploring resources like the Japan Biobank, which has sex-stratified GWAS, or other biobank studies.

While identifying potentially causal genes is beneficial, their biological relevance should be discussed with consideration of sex differences. The impact of candidate genes on phenotypic variance between sexes hasn't been comprehensively addressed in the main text. Additionally, I recommend releasing the data from this new analysis to prioritize gene candidates.

The author presents the equality test, offering intriguing insights into sex differences in genetics. To enhance understanding, it would be beneficial to provide more biological interpretation and examples of how the proposed model can elucidate these differences.

(Remarks on code availability)

Reviewer #2

(Remarks to the Author)

Thanks to the authors for addressing my comments!

I have one follow-up question: Given the female sampling bias in the UK Biobank and the importance of validating findings in a replication cohort, can the authors try to replicate the loci found in UKBB with sex differences in another biobank?

(Remarks on code availability)

Reviewer #3

(Remarks to the Author)

The authors have addressed the comments appropriately. More specifically, I think the additional text and simulation analyses have improved the readability and value of the manuscript. I still think that the identification of a positive control for females should be possible, and though I appreciate that it is harder, I think more could be done to identify this and run appropriate additional analyses. Nonetheless, overall, I think this manuscript does indeed add to the important field of sex differences in genetic traits and I would recommend it for publication.

(Remarks on code availability)

Reviewer #4

(Remarks to the Author)

(Remarks on code availability)

Version 2:

Reviewer comments:

Reviewer #1

(Remarks to the Author)

My concerns have been addressed.

(Remarks on code availability)

REVIEWER COMMENTS

Reviewer #1 (Remarks to the Author):

In this manuscript, Uffelmann et al. reported local sex differences in the genetic effects of 157 quantitative traits in the UK Biobank. They investigated sex differences in local heritability (h^2), genetic correlation (r_g), as well as the absolute differential effective size of r_g . Traits that show greater differences are phenotypes that are known to be sex dimorphism. Although sex differences in genetic effects at a regional scale have been sparsely studied, some of the discoveries in the manuscript have been previously reported in the same population (Elena et al., Nature Genetics, 2021). In fact, the genome-wide genetic effect comparison in the Nat Gen paper provides more biological insights than this manuscript, as it identifies specific SNPs/genes rather than regions, which may encompass multiple genes. Another question regarding this manuscript is the interpretation and validation of their findings. Whether the G x Sex differences could be causal to the observed sex differences in the traits, and the biological mechanisms underlying such differences, have not been comprehensively illustrated. Only a few known examples have been provided. Also, the manuscript lacks discussion on the limitations of their study and the comparison of their results to other relevant publications.

We thank the reviewer for their comments. To address them, we have made the following changes:

- We have added a gene-level analysis for the equality of genetic effects and equal heritabilities.
- We have added FLAMES analyses to predict the most likely causal gene in a locus, addressing whether genes with GxS effects are potentially causal for the traits.
- We have added more analyses and discussions on potential biological mechanisms underlying the observed sex differences.
- We expanded the discussion to list the potential limitations of our study.

We provide more detailed replies to the comments below.

Here are some detailed comments:

1. Lacking interpretation of those sex differential regional genetic effects. Sex differences in h^2 and r_g in a region indicate that the genetic contribution to the trait differs between sexes in the given region. However, since each region may contain many genes and regulatory elements, the observed sex differences in r_g and h^2 do not provide further insight into causal discovery or biological mechanisms. It is uncertain how to interpret these differences, such as identifying the putative causal gene for the trait.

In response to this reviewer's remarks, we have now applied FLAMES², a gene-prioritization method, to identify putatively causal genes and test for equal heritabilities and equal genetic effects specifically for these genes of interest. See pages 13-15:

Because the equality test can be applied to arbitrarily small loci, we repeated the analysis using gene boundaries as locus definitions to improve their biological interpretation. However, we note that applying it to smaller loci while increasing the Bonferroni correction using the number of genes (i.e. $0.05 / (18576 \times 157) = 1.7e-08$) comes at the expense of power. Of all 157 traits, 55 had at least one gene that significantly differed between males and females on at least one scale. Of these 55 traits, the median number of significant genes was 2. The correlation of p -values between equality_{raw} and equality_{std} across all traits and genes was 0.98, likely because

most genes did not differ between the sexes on either scale and because the phenotypic variances were mostly similar (mean female-to-male ratio: $0.95 \pm \text{SE } 0.03$), except for testosterone (female-to-male ratio: 0.03) and oestradiol (female-to-male ratio: 36.42). However, we identified 85 genes across 10 traits with divergent results in the $\text{equality}_{\text{raw}}$ and $\text{equality}_{\text{std}}$ test (i.e. one test was significant at $p < 1.7\text{e-}08$, while the other had $p > 0.01$). For Sex Hormone Binding Globulin (SHBG), three neighboring genes (i.e. NRBF2, JMJD1C, and REEP3) showed very similar and strong sex differences with $\text{equality}_{\text{std}}$ ($p < 1.3\text{e-}34$), but not with $\text{equality}_{\text{raw}}$ ($p \geq 0.7$). The phenotypic variance of SHBG was 3.6 times larger in females, therefore the heritability was expected to be larger in males (see column 2 in Figure 5). Indeed, the heritability was significantly different ($p < 1\text{e-}08$) and more than 3 times larger in males for all three genes ($h^2_{\text{local, females}} = 0.3\%$, $h^2_{\text{local, males}} = 1.1\%$). JMJD1C has been hypothesized to impact SHBG levels via thyroid hormones that affect HNF4A²², and while this seems to apply to both males and females, our results suggest this to be a much stronger effect in males. Reversely, for HDL cholesterol, the gene CETP showed strong sex differences with $\text{equality}_{\text{raw}}$ ($p = 3.62\text{e-}32$) but not with $\text{equality}_{\text{std}}$ ($p = 0.02$). As expected (see column 3 in Figure 5), the heritabilities did not significantly differ ($p = 0.07$, $h^2_{\text{females}} = 4.6\%$, $h^2_{\text{males}} = 4.3\%$). The phenotypic variance in females was 1.5 times larger than in males which explains the observed difference with $\text{equality}_{\text{raw}}$.

Next, we tested whether significant loci in the equality test can highlight known biological differences between the sexes. As a positive control, we highlighted results for testosterone because its biology is well-understood and known to be markedly different between males and females^{13,16}. Whether or not a gene has a known function in testosterone biology strongly predicted association strength in the equality test ($p = 8.13\text{e-}09$). Reversely, nearly every significant locus in the gene-based equality test highlighted known biological differences in testosterone biology (see Figure 6). Most testosterone GWAS hits did not overlap between males and females¹⁶, but differences may exist even for those that do. For example, we found AKR1C3, a gene associated with testosterone synthesis and metabolism, to map to a GWAS hit in males (10:5118231:5356763) and females (10:5026680:5114693). However, the lead SNPs are on opposite sides of AKR1C3, and both $\text{equality}_{\text{raw}}$ ($p = 5.19\text{e-}12$) and $\text{equality}_{\text{std}}$ ($p = 1.52\text{e-}14$) showed significant differences for this gene. There was no difference in h^2_{local} ($p = 0.24$), but the 1Mb locus that contained AKR1C3 had a $r_{\text{g, local}}$ of 0.18, which was significantly different from 1 ($p < 1\text{e-}8$) but not from 0 ($p = 0.17$). As such, while AKR1C3 contains signals in both males and females, the pattern of SNP associations is markedly different. Moreover, the gene prioritization tool FLAMES²³ predicted different causal genes at this locus, namely AKR1C2 for females and AKR1C4 for males, which we also found to be significantly different in the equality test (see Figure 6).

For diastolic blood pressure, we replicated a recent study that found the COL4A1/COL4A2 locus to be sexually dimorphic²⁴ (see Figure 6). This locus is significant in the male, but not the female GWAS. FLAMES²³ predicted COL4A1 to be the most likely causal gene for males. Coincidentally, the equality test flags this gene to differ the most on both scales. However, we do not find evidence for differential effects of three other loci (i.e. PECAM1, NT5C2, MSTN) identified in the same study²⁴. These loci did not reach statistical significance in an additional sex-by-genotype interaction analysis in the same study²⁴, and may thus be false positives.

Lastly, APOE, which FLAMES predicted to be the causal gene at a GWAS hit for low-density lipoprotein (LDL) levels in both males and females, shows a pronounced sex difference with both $\text{equality}_{\text{raw}}$ ($p = 6.78\text{e-}81$) and $\text{equality}_{\text{std}}$ ($p = 7.60\text{e-}82$) (see Figure 6). While the $r_{\text{g, local}}$ of the locus (19:45040933:45893307) is 0.996, the h^2_{local} of APOE, while large in both, is twice larger in females ($h^2_{\text{local}} = 6\%$) than in males ($h^2_{\text{local}} = 3\%$). Similarly, we found APOE to be sexually dimorphic for high-density lipoprotein (HDL) levels with both $\text{equality}_{\text{raw}}$ ($p = 7.39\text{e-}17$) and $\text{equality}_{\text{std}}$ ($p = 1.49\text{e-}13$).

2. Limited in novel discoveries. In the manuscript, the author provided several examples of literature-reported evidence to support their discovery. While this serves as proof of concept, there is a lack of additional functional annotation for other discoveries, except for testosterone, which has been extensively studied before (Nat Med 26, 252–258 (2020)). Additional analysis is needed to characterize novel discoveries and improve the validity and impact of this work.

We have now added functional annotations for other discoveries, including SHBG levels, diastolic blood pressure, LDL, and HDL levels.

Specifically:

1. JMJD1C's male-biased effect on SHBG levels (page 13):

For Sex Hormone Binding Globulin (SHBG), three neighboring genes (i.e. NRBF2, JMJD1C, and REEP3) showed very similar and strong sex differences with equality_{std} ($p < 1.3e-34$), but not with equality_{raw} ($p \geq 0.7$). The phenotypic variance of SHBG was 3.6 times larger in females, therefore the heritability was expected to be larger in males (see column 2 in Figure 5). Indeed, the heritability was significantly different ($p < 1e-08$) and more than 3 times larger in males for all three genes ($h^2_{local, females} = 0.3\%$, $h^2_{local, males} = 1.1\%$). JMJD1C has been hypothesized to impact SHBG levels via thyroid hormones that affect HNF4A³, and while this seems to apply to both males and females, our results suggest this to be a much stronger effect in males.

2. A recent study⁶ identified 4 loci to be sexually dimorphic for diastolic blood pressure. However, only 1, namely COL4A1/COL4A2, remained significant in a subsequent genotype-by-sex interaction analysis. We set out to investigate this closer and only found evidence for COL4A1/COL4A2, but not the other postulated loci to be sexually dimorphic (page 14):

For diastolic blood pressure, we replicated a recent study that found the COL4A1/COL4A2 locus to be sexually dimorphic⁶ (see Figure 6). This locus is significant in the male, but not the female GWAS. FLAMES² predicted COL4A1 to be the most likely causal gene for males. Coincidentally, the equality test flags this gene to differ the most on both scales. However, we do not find evidence for differential effects of 3 other loci (i.e. PECAM1, NT5C2, MSTN) identified in the same study⁶. These loci did not reach statistical significance in an additional sex-by-genotype interaction analysis in the same study⁶, and may thus be false positives.

3. We extended our analyses of sex differences of APOE in LDL-direct and HDL (pages 14-15):

Lastly, APOE, which FLAMES predicted to be the causal gene at a GWAS locus for low-density lipoprotein (LDL) levels in both males and females, shows a pronounced sex difference with both equality_{raw} ($p = 6.78e-81$) and equality_{std} ($p = 7.60e-82$) (see Figure 6). While the $r_{g, local}$ of the region (19:45040933:45893307) is 0.996, the h^2_{local} of APOE, while large in both, is twice larger in females ($h^2_{local} = 6\%$) than in males ($h^2_{local} = 3\%$). Similarly, we found APOE to be sexually dimorphic for high-density lipoprotein (HDL) levels with both equality_{raw} ($p = 7.39e-17$) and equality_{std} ($p = 1.49e-13$).

3. The rationale for choosing the h^2 threshold. The $h^2 p < 1e-4$ was used as the region selection criterion for further analysis. It is uncertain whether there are any genome-wide significant loci in the region or if any genome-wide significant loci would be missed by using this cutoff. To better define the rationale for choosing this cutoff, it is necessary to test how many credible sites

under different cutoffs in a region should be examined. In fact, given that quantitative traits were investigated here, it is arguable that a more stringent cutoff should be used to avoid false positive discoveries, as continuous traits have greater power and often yield stronger GWAS signals than binary traits.

The purpose of this filtering step is to remove regions with little genetic variance that are not worth further analysis. This is because genetic correlations cannot exist if there is no genetic variance to begin with. Applying a threshold, such as $1.00e-4$, improves computational efficiency.

We want to note that the h^2 filtering threshold of $p < 1.00e-4$ filters out regions with low evidence for genetic signal and does not determine the false positive ratio of the genetic correlations, which have their own significance threshold. The p-value threshold for the correlations is corrected for the total number of bivariate tests performed (i.e. $0.05 / 11259 = 4.44e-06$) and is thus indirectly influenced by the threshold of $p = 1.00e-4$ as that determines how many regions are tested. Using a *more stringent* h^2 p-value filtering threshold would lead to *more* regions being excluded from testing genetic correlations, and thus fewer tests performed resulting in a more lenient significance threshold for the genetic correlations. If we would make the filtering threshold more lenient and for example include all regions, that would result in a more stringent Bonferroni correction on the genetic correlation, but would also be computationally very heavy with a lot of regions included that have no genetic signal. Therefore we opted for the current 2-step process – first select regions with evidence for genetic signal ($p < 1.00e-4$) then conduct bivariate tests across males and females.

We have added a table to the supplement, summarizing the total number of loci and the number of loci with genome-wide significant SNPs at different h^2 p-value thresholds. As shown in the table, setting the h^2 p-value threshold to $1e-4$ reduces the total number of loci from 391,715 to 27,989 (93% reduction of loci to be tested) while retaining 82% (9573 / 11661) of loci with genome-wide significant SNPs in females. The percentages are nearly the same in males.

We have added clarification to the methods (page 21):

All LAVA analyses were performed on 2495 semi-linkage disequilibrium independent blocks of approximately 1Mb in size (<https://github.com/josefin-werme/LAVA>). Local genetic correlations were only estimated for loci with sufficient evidence of heritability in both males and females. This is because a genetic correlation cannot exist if no genetic variance is present, and filtering loci with little to no heritability improves computational efficiency while reducing the multiple testing burden. To this end, we applied a h^2_{local} threshold of $p < 1.00e-04$. Using this threshold reduced the number of loci by 93% while retaining 82% of loci with genome-wide significant SNPs in both males and females (see Supplementary Table 3).

4. What is the interpretation of a negative rg ? Does this imply an opposite genetic effect of the region? Does this imply a potential differential environmental effect on the trait between sexes?

A negative genetic correlation implies that the pattern of underlying genetic effects goes in opposite directions. However, the underlying model that gives rise to such a correlation is not postulated. For an environmental effect to induce a negative genetic correlation, it would have to directly modulate the effects of genetic variants oppositely for men and women. While being an interesting thought experiment that may warrant further analysis and simulations, it is beyond the scope of the current study.

Additionally, what does the statement in line 91, " rg is significantly different from zero," mean?

A genetic correlation can, in principle, be tested against any null hypothesis of interest (e.g. $rg = 1$, $rg = 0$, or $rg = 0.25$). Our study primarily tests whether the observed correlations are

different from 1 (i.e., the null hypothesis is that the genetic effects are perfectly linearly correlated). However, for regions significantly different from 1, it may be interesting to test whether they also differ from 0. If they do not, we cannot reject the null hypothesis that there is no linear relationship between the genetic effects of males and females at all. The p-values provided by the standard LAVA bivariate analyses use the null hypothesis of $r_g = 0$, as described in ⁷, because the most common use of LAVA is to test whether an r_g exists between two different traits. In the present study however, we were mostly interested in testing whether the r_g between males and females deviated from 1. We have added the following to the Methods section (page 24):

To test the null hypothesis $H_0: \rho_{pq} = \rho_0$ of a perfect local genetic correlation (where $\rho_0 = 1$) between two phenotypes, we generalized the base LAVA model (which tests $H_0: \rho_{pq} = 0$ by default)

5. In the testosterone example, genes in regions were selected based on existing knowledge. However, whether the genes actually lie within the causal locus is unknown. This highlights the main limitation that regional test is limited in identifying putative causal genes/loci.

Indeed, the 1Mb-sized regions are defined independently of genetic association evidence, and thus may or may not overlap with GWAS loci. However, we address this remark by using FLAMES to indicate putative causal genes in GWAS loci and applying the test of equal genetic effects (*equality_{raw}* and *equality_{std}*) and the test of equal heritabilities to these genes, in addition to filtering out genomic regions where there was low evidence for genetic signal. We kindly refer to our answer to comment #1, where we discuss this in more detail.

Reviewer #2 (Remarks to the Author):

Uffelmann and colleagues present a study to investigate the local genetic differences between males and females in 157 quantitative traits using the publicly available GWAS summary statistics in UKBB. They compared local heritability estimates, estimated local genetic correlation, and tested the equality of genetic effects between sexes. They found that most of the quantitative traits that were studied have at least one locus that shows significant difference in males and females. They highlighted several examples, including loci for testosterone, the locus containing the gene MCPT2 for Cystatin C and the locus containing APOE for LDL-direct. While the results are interesting, there are some major issues that I would like to bring to the authors' attention.

Several sections within the manuscript were not presented with clarity.

We thank the reviewer for their comments. To address them, we have made the following main changes:

- We have added simulations to model power differences due to the female sampling bias in the UK Biobank, as well as to evaluate type-1 error rates of the newly developed tests
- We have updated the Testosterone analysis to provide more clarity. That is, we removed the MAGMA analysis and replaced it with gene-level results from the equality of genetic effects test
- We added analyses on the overlap between loci with significant sex differences and significant GWAS hits

We provide more detailed replies to the comments below.

1. The authors analyzed 157 traits in the UKBB, it is unclear how those quantitative traits were selected. For example, in the methods (Line 177), it reads "Additionally, phenotypes were

removed that were suspected to have minimal genetic basis (e.g. 'age at recruitment').“, What metrics were used here? Were global heritability estimates used for filtering?

Thank you for bringing our attention to this lack of clarity. We first considered all traits that were available in the Nealelab and had not been transformed before running the GWAS (i.e., our analyses require the GWAS to have been run on the raw phenotypic scores). We manually removed traits, as the reviewer correctly notes, that we suspected to have minimal genetic bases based on the trait names (“age at recruitment”). Similarly, we excluded duplicated traits (e.g., impedance measures of weight vs. other measures). To add more transparency, we added a list of the traits we considered and whether or not we included them in our analyses (Supplementary Table 1). Moreover, we note in the Methods that we did not apply any quantitative thresholds to exclude traits (page 20):

Summary statistics for 157 traits (see Supplementary Table 1) were downloaded from the Nealelab (https://github.com/Nealelab/UK_Biobank_GWAS). Only summary statistics for untransformed (i.e., raw) and quantitative phenotypes were downloaded. Phenotypes that were deemed too similar were filtered out, such that only one phenotype remained (e.g., hand grip strength left vs. right; see Supplementary Table 1 for a full list of traits that were considered and those that have been selected). Additionally, phenotypes that were suspected of having a minimal genetic basis were removed (e.g., 'age at recruitment').

2. Similarly, it would be great to have clear cutoffs for excluding or including trait and loci in analyses. For example, in Line 44, it reads “After the exclusion of loci with very large h^2_{local} s (i.e. lipoprotein A, direct bilirubin, and total bilirubin)”. Please clarify which cutoff for heritability was used.

Thanks for pointing out this omission. We removed loci with $h^2_{\text{local}} > 0.2$. This led to the removal of the three loci. We have added this to the text and a supplementary figure of the scatterplot before and after applying the filter (page 5):

Across all traits, h^2_{local} s for loci that were significant in both males and females correlated strongly ($r = 0.98$, $p < 1.00\text{e-}300$). After the exclusion of loci with very large h^2_{local} s (i.e. > 0.2 , for lipoprotein A, direct bilirubin, and total bilirubin; see Supplementary Figure 1), the correlation estimate decreased but remained high ($r = 0.89$, $p < 1.00\text{e-}300$).

3. Line 84, it reads “For many traits with LDSC- r_{g} , global close to one, we estimate some r_{g} , local that are closer to zero and in some instances negative.”. Please clarify the specific number of traits before describing examples.

We have added the exact numbers to the text as follows (page 9):

Out of 119 traits with LDSC- $r_{\text{g, global}}$ not significantly different from one, 84 had at least one $r_{\text{g, local}}$ that was significantly different from 1.

4. The authors mentioned the power difference in females and males due to the female sampling bias in the UK Biobank and GWAS in females tend to have larger sample sizes than in males (Line 51). Please provide more descriptions about the sample size differences including which traits have the largest differences.

We have added more information to the text and, in addition, added Supplementary Figure 2, plotting male and female sample sizes against each other while highlighting traits with the

largest difference. Moreover, we have added simulations to study the effect of the sample size asymmetry. While we did find power asymmetries at extreme sample size differences between males and females (sample size ratio ≥ 2), we did not find any notable asymmetries at sample size ratios typical of the traits we study (median sample size ratio = 1.16). Nonetheless, we note in the manuscript that when sample sizes start to diverge more strongly, there is more power to detect heritability differences when the sex with the smaller sample size has the larger heritability. See Supplementary Figure 3 and pages 5-6:

Approximately 62% of all significant h^2_{local} s are significant in only one of the sexes. Across all traits, we identify more loci with significant h^2_{local} s in females than males (10614 loci across 153 traits vs. 8767 loci across 151 traits). This is at least partly due to the sex-stratified GWASs' sample sizes being mostly larger in females, increasing the power of h^2_{local} analyses in females relative to males. Specifically, 153 out of 157 traits have larger sample sizes in females with a median female:male sample size ratio of 1.16 (see Supplementary Figure 2 and Supplementary Table 1 for exact sample sizes). This is a consequence of the female sampling bias in the UK Biobank. Simulations show that this can result in power asymmetries at large sample size differences (see Supplementary Figure 3 and Methods), with somewhat greater power to detect differences in heritability if the larger heritability is for the sex with the smaller sample, rather than the other way around. However, this power asymmetry is largely negligible for the sample size differences typical of the UK Biobank.

5. Line 62, for locus and trait to have sufficient genetic signals, besides p-values for h^2_{local} , the magnitude is also important.

It is correct that the magnitude of SNP effect sizes affects the amount of genetic signal, or heritability, in a locus. However, all else being equal, if the effect sizes increase, the p-values for the corresponding local heritability estimates will also decrease (i.e., become more significant). Therefore, filtering on the p-values for the local heritability estimates should be sufficient in this context.

(We did not know whether this comment was a question or a remark; we agree with the remark but are unsure whether any changes are needed in the manuscript.)

6. Can authors use the equality test to disentangle the reasons for loci to have different h^2_{local} ? Do those loci have different effect sizes or allele frequencies between sexes? For example, the locus 2181 (17:7264459:8554763) for testosterone was found to be significant in the equality test. What are the allele frequencies in males and females for this locus?

No known biological mechanism can lead to autosomal allele frequency differences between men and women. This is due to Mendel's law of random segregation, which is independent of the sex chromosomes. Previous analyses of allele frequencies in men and women have either found no difference⁸ or have studied them in the context of participation bias⁹ (we note the results and conclusions of the latter have been contested, and thus it remains a controversial topic¹⁰). We have added this to the introduction of the manuscript (page 3):

Male and female autosomal allele frequencies do not differ due to Mendel's law of segregation, which is independent of the sex chromosomes. Therefore, genetic sex differences are expected to manifest biologically only in causal variant effects.

The equality test can indeed be combined with the test of equal heritabilities and genetic correlations to shed more light on why we observe certain differences between the sexes. We have added a new main figure to aid with the interpretation of the equality test (Figure 5). Locus 2181 for Testosterone has indeed different heritabilities in males (4.3%) and in females (0.06%), while the local genetic correlation is not significantly different from 1. The equality test

is significant on both the raw phenotypic and the standardized scales. The phenotypic variance in males is ~ 35 times larger than in females. This scenario reflects column 4 in Figure 5. While the phenotypic variance is much larger in males, the genetic effects do not simply scale with this but are proportionally even larger, which, in turn, results in a much larger heritability in males. In this way, combining and interpreting these different tests and the phenotypic variances together allows for a detailed characterization of genetic sex differences. We have added several examples like these to the text, for example (page 13):

For Sex Hormone Binding Globulin (SHBG), three neighboring genes (i.e. NRBF2, JMJD1C, and REEP3) showed very similar and strong sex differences with equality_{std} ($p < 1.3e-34$), but not with equality_{raw} ($p \geq 0.7$). The phenotypic variance of SHBG was 3.6 times larger in females, therefore the heritability was expected to be larger in males (see column 2 in Figure 5). Indeed, the heritability was significantly different ($p < 1e-08$) and more than 3 times larger in males for all three genes ($h^2_{local, females} = 0.3\%$, $h^2_{local, males} = 1.1\%$). JMJD1C has been hypothesized to impact SHBG levels via thyroid hormones that affect HNF4A³, and while this seems to apply to both males and females, our results suggest this to be a much stronger effect in males. Reversely, for HDL cholesterol, the gene CETP showed strong sex differences with equality_{raw} ($p = 3.62e-32$) but not with equality_{std} ($p = 0.02$). As expected (see column 3 in Figure 5), the heritabilities did not significantly differ ($p = 0.07$, $h^2_{females} = 4.6\%$, $h^2_{males} = 4.3\%$). The phenotypic variance in females was 1.5 times larger than in males which explains the observed difference with equality_{raw}.

7. To understand the biological differences between the sexes, MAGMA gene analysis was performed. The description is unclear why the gene analysis was used here. How the co-localization was performed to connect the genes identified by MAGMA that are significantly associated in males and females separately to the equality tests. The authors described the pathways, to which genes are related. The results could be biased without pathway enrichment analyses.

We agree that the combination of the MAGMA gene analysis and the equality test in one plot was unclear and could lead to confusion. Therefore, we have decided to replace the plot with one only showing the equality test p-values per gene and to remove the MAGMA analysis. That is, we have repeated the equality test but applied it to single genes, instead of 1Mb-sized LD regions. We hope that this will improve the interpretation of the results. We have plotted the results in a Miami plot in Figure 6, where each point represents the p-value from the equality test on the raw (top) and standardized (bottom) scales. For testosterone, we annotated genes of known function in testosterone biology that also reached Bonferroni-corrected significance on at least one of the two scales. This illustrates how most significant loci in the equality test harbor genes of known function in testosterone. Moreover, whether or not a gene is known to have a function in testosterone biology strongly predicts its equality test p-value ($p = 8.13e-09$). We have changed the manuscript accordingly (pages 13-14):

Next, we tested whether significant loci in the equality test can highlight known biological differences between the sexes. As a positive control, we highlighted results for testosterone because its biology is well-understood and known to be markedly different between males and females. Whether or not a gene has a known function in testosterone biology strongly predicted association strength in the equality test ($p = 8.13e-09$). Reversely, nearly every significant locus in the gene-based equality test highlighted known biological differences in testosterone biology (see Figure 6). Most testosterone GWAS loci did not overlap between males and females³, but differences may exist even for those that do. For example, we found AKR1C3, a gene associated with testosterone synthesis and metabolism, to map to a GWAS locus in males (10:5118231:5356763) and females (10:5026680:5114693). However, the lead SNPs are on opposite sides of AKR1C3, and both equality_{raw} ($p = 5.19e-12$) and equality_{std} ($p = 1.52e-14$) showed significant differences for this gene. There was no difference in h^2_{local}

($p = 0.24$), but the 1Mb region that contained AKR1C3 had a $r_{g, local}$ of 0.18, which was significantly different from 1 ($p < 1e-8$) but not from 0 ($p = 0.17$). As such, while AKR1C3 contains signals in both males and females, the pattern of SNP associations is markedly different. Moreover, the gene prioritization tool FLAMES¹ predicted different causal genes at this locus, namely AKR1C2 for females and AKR1C4 for males, which we also found to be significantly different in the equality test (see Figure 6).

8. The $r_{g, local}$ and h^2_{local} were estimated in this study in the LD independent blocks across the genome. How many of the loci identified to have differences in females and males are GWAS loci?

We have added a breakdown of the number of GWAS loci overlapping with regions that significantly differed between males and females in the equality test. The numbers are given for the equality test because any region with significantly different correlations or heritabilities will also tend to be significant in the equality test. Therefore, it provides the largest set of sex-dimorphic regions. Pages 12-13:

Not all significant loci in the equality tests overlap with GWAS loci. Out of 606 significant equality_{raw} loci, 89 were significant GWAS loci in females and 130 in males. For equality_{std}, 64 female and 69 male GWAS loci overlapped with 512 regions that significantly differed between males and females.

Moreover, we have added this as a limitation to the Discussion on pages 18-19:

Fourth, loci that are significantly different between males and females do not necessarily contain significant GWAS hits. As such, these loci may contribute only minimally to phenotypic differences. Heritability estimates of the given locus can be evaluated to mitigate this, or loci may be filtered to those containing genome-wide significant SNPs.

9. The authors extended LAVA to test for equality of genetic effects. Please evaluate the performance of the test, especially for the type I error rates or FDR using simulations

We have added simulation results of type-1 error rates for each test to the manuscript. A table with numeric error rates can be found in Supplementary Table 2, and a description of the simulations can be found in the Methods (pages 25-26):

Type-1 error simulations

Type 1 error simulations were performed by first generating 250 independent principal components W for the desired sample size (with 250 being generally representative of the number of genetic principal components per block in the LAVA analyses). Continuous phenotypes were then simulated by first setting the true (raw scale) genetic effect vectors δ_p^* and δ_q^* for each of the two phenotypes p and q according to the desired null model, computing the genetic components $G_p = W\delta_p^*$ and $G_q = W\delta_q^*$, and adding normally distributed noise to these with variance set to obtain a specified phenotypic variance to obtain simulated phenotypes Y_p and Y_q . These were each standardized and then regressed on W to obtain the estimates $\hat{\delta}_p$ and $\hat{\delta}_q$ of the standardized effects $\delta_p = \frac{\delta_p^*}{SD(Y_p)}$ and $\delta_q = \frac{\delta_q^*}{SD(Y_q)}$, which were then used as input for the test being evaluated.

For each condition, 10,000 repeats were generated, and type 1 error rates were computed for α values of 0.05 and 0.001 as the proportion of repeats for which $p < \alpha$. For each of the four evaluated tests, for the first phenotype the variance $\text{var}(Y_p)$ was always set to one and the heritability h_p^2 was varied across 0%, 0.1%, 0.5% and 1%. Sample sizes were set to either 10,000 or 50,000.

For the test of equality of genetic effects, simulations were performed for equality under the raw scale ($H_0: \delta_p^* = \delta_q^*$) as well as the standardized scale ($H_0: \delta_p = \delta_q$), with the variance of the second phenotype was set either to $\text{var}(Y_q) = \text{var}(Y_p) = 1$ or to $\text{var}(Y_q) = 2 \times \text{var}(Y_p)$. For the test of equality of local heritabilities, the null model was set to $H_0: h_p^2 = h_q^2$, with $\delta_p \neq \delta_q$ and $\text{var}(Y_q) = \text{var}(Y_p) = 1$. Finally, for the test of perfect local genetic correlations, the null model $H_0: \text{cor}(G_p, G_q) = 1$ was used. This is equivalent to $H_0: \delta_q = c\delta_p$ for an arbitrary positive value of c , which implies $h_q^2 = c^2 h_p^2$. The parameter c was set to either 1 or $\sqrt{2}$, and $\text{var}(Y_q) = \text{var}(Y_p) = 1$. For these simulations the $h_p^2 = 0$ conditions were omitted, as the local genetic correlation is not defined if no local genetic variance is present for either of the phenotypes. Exact type-1 error rates for each test and condition can be found in Supplementary Table 2.

The test of equality of genetic effects (both scales) and the test of equal heritabilities were well-controlled across all conditions. We found the test of perfect correlation of local genetic signal to be slightly deflated at small sample sizes ($N \sim 10,000$). However, it's well controlled at larger sample sizes and heritabilities. In any case, none of the tests have type-1 error rates that are elevated, and therefore, false discoveries are unlikely to occur.

10. Because LAVA can also be used for binary phenotypes, can authors discuss why only quantitative traits were studied in this paper? Are there limitations in the current analyses so that binary phenotypes are not included?

At present, the additional LAVA tests employed here have only been validated for continuous phenotypes, work to validate these tests for use with binary phenotypes is currently ongoing. We have clarified this in the paper for each test in the methods section (pages 23,24 and 25)

| We note that, at present, this test has only been evaluated for quantitative phenotypes.

and have added it as a limitation to the Discussion (page 18):

| Second, we have only considered quantitative traits as work validating the newly developed tests for binary traits is ongoing.

Minor:

- Please cite LAVA at its initial appearance in the manuscript (Line 89)

We now cite LAVA at its first appearance (page 5):

Local heritability.

The autosomal genome was divided into 2495 semi Linkage Disequilibrium (LD)-independent regions of approximately 1Mb in size (see Methods and ref⁶). We computed local heritability estimates (h_{local}^2) for all traits with LAVA⁶.

- In Line 45, is the Pearson's correlation estimate reported? Please clarify.

We used Pearson correlation and have clarified this in the manuscript (page 5):

| Across all traits, h_{local}^2 for loci that were significant in both males and females correlated strongly ($r_{\text{pearson}} = 0.98, p < 1.00\text{e-}300$). After the exclusion of loci with very large h_{local}^2 (i.e.,

lipoprotein A, direct bilirubin, and total bilirubin with $h^2_{\text{locals}} > 0.2$; see Supplementary Figure 1), the correlation estimate decreased but remained high ($r_{\text{pearson}} = 0.89, p < 1.00e-300$).

Reviewer #3 (Remarks to the Author):

The manuscript “Local Genetic Sex Differences in Quantitative Traits” addresses an important area of genetic research, namely sex differences in the heritability of quantitative traits, for the first time looking at local heritability. Given the increasing focus on precision medicine, together with sample sizes that enable sex-stratified analyses, this is a timely piece of work that attempts to fill a genuine gap and which I think will be of broad interest. The authors find that while most loci do not differ in terms of direction or magnitude of effect across the sexes, almost every trait that was studied (157 quantitative traits) did have at least one locus that did show a difference. Overall, the results are interesting but there are significant limitations to this study.

We thank the reviewer for their comments. To address them, we have made the following main changes:

- We have expanded the manuscript, particularly the introduction and discussion, to provide more background information and to integrate it more thoroughly with the existing literature
- We have added Supplementary Figures 5 & 6 to study the correlations between all 157 traits we considered in our study
- We have run simulations to evaluate the effect of female-biased sample size ratios.

We provide more detailed replies to the comments below.

General comments

It is not clear to me why the authors have chosen to only study sex differences in quantitative traits.

At present, the additional tests that were developed for the current manuscript have only been validated for continuous phenotypes; work to validate these tests for use with binary phenotypes is currently ongoing. We have clarified this in the paper for each test in the methods section (pages 23, 24 and 25)

| We note that, at present, this test has only been evaluated for quantitative phenotypes.

and have added it as a limitation to the Discussion (page 18):

| Second, we have only considered quantitative traits as work validating the newly developed tests for binary traits is ongoing.

It also unclear, why only sex stratified GWAS analyses using UK Biobank data have been used. This should be explained.

UK Biobank is currently the only large resource for which sex-stratified summary statistics for a large number of traits are already available. Although other biobanks exist, gaining access and running sex-stratified GWAS from scratch for hundreds of traits would take a significant amount of additional computing time. Since our purpose was to provide a broad overview of the presence of local genetic differences between males and females and since the UK Biobank sample is well-powered (337,199 individuals of British ancestry), we decided to only use sex-stratified GWAS from the UK Biobank. We have now clarified this in the limitations (page 18):

Third, we only considered data from the UK Biobank. This is because it is the only large biobank we are aware of for which sex-stratified GWAS summary statistics have been computed and made publicly available for hundreds of traits.

Furthermore, the manuscript is very short given the complexity of the topic with a particularly short introduction and discussion. This means that readers without a background in the field would find it hard to contextualise the work, or fully understand the results and methods. Therefore, I think it would be sensible for the authors to extend the text, particularly the introduction and discussion.

We have now expanded the manuscript considerably, particularly in the Introduction and Discussion. We have also added subheadings and more material. We will not copy all of these additions here, but kindly refer the reviewer to all textual changes in the manuscript, which are highlighted in yellow.

Finally, only limited effort is made to dissect or understand the molecular basis of sex differences in local heritability and the implications for diagnostics or treatment e.g. could such findings point to the need for sex-stratified PRS or potentially explain sex differences in response to treatment. While I appreciate that these questions may be beyond the scope of the manuscript itself, they could be considered and discussed.

We agree that these are important questions that require studies in their own right. We have added implications of our study for the potential of sex-stratified PRS analyses (page 18):

While nearly every trait showed some evidence for localized genetic sex differences, most regions for most traits did not differ. As such, while not tested in this study, we expect that genome-wide and additive polygenic predictions based on sex-stratified GWASs will not outperform those based on sex-combined GWASs. This is because sample sizes would be halved, thus drastically reducing power. However, other approaches that directly model GxS and polygenic covariance structures between males and females can improve risk prediction, particularly for traits with genome-wide sex differences such as testosterone and waist-to-hip ratio¹⁰.

We feel reluctant to discuss implications for diagnostics or treatments at length as we have not studied disease traits. However, we tentatively refer the reviewer to our discussion of the APOE locus that we found to be sexually dimorphic for LDL and HDL. See page 18:

Combining LAVA bivariate $r_{g, \text{local}}$ and univariate h^2_{local} analyses with the test of equal heritabilities, perfect $r_{g, \text{local}}$ and equality of genetic effects allows for very detailed descriptions of local genetic sex differences. In this way, for LDL levels, we found APOE to have different magnitudes of genetic effects on both scales while the $r_{g, \text{local}}$ was not different from one, leading to h^2_{local} estimates that are twice as large in females. Future studies could compare whether and how this relates to the sex-dimorphic effects of APOE on Alzheimer's Disease¹¹⁻¹⁵. Moreover, we show for the AKR1C testosterone GWAS locus, how a $r_{g, \text{local}}$ of zero and significant equality tests can point to two different causal genes for males and females. In the future, this type of analysis may be applied to disease traits to test for genetic sex differences that may point to different causal pathways. This, in turn, may inform sex-specific treatments.

Specific comments

Focusing on the first analysis presented (Fig. 1)

- The autosomal genome is divided into 2495 'semi LD independent' loci of ~1Mb with h^2_{local} computed between all traits and all loci. While this method is referenced (Werne et al., 2022:

‘An integrated framework for local genetic correlation analysis’) to understanding exactly how the partitioning was performed required reading supplementary methods of the Werne et al. paper. Given the importance of partitioning for local heritability analyses, it would be helpful for this information to be easier to find and the rationale explained.

We agree that the genome partitioning is central to our manuscript. Therefore, we have added a description to the Methods (pages 20-21):

Genome partitioning

The genome was partitioned into blocks in the same way as in ⁶, the method for which is described there in full. In brief, this partitioning method aims to divide the genome into smaller blocks of roughly equal size while minimizing the level of LD between them. It does so by recursively splitting the genome (starting at whole chromosomes), each time selecting the breakpoint for which the local LD between SNPs across that break point is lowest. This process is repeated until no further valid breakpoints can be found; a breakpoint is invalid if it results in blocks containing fewer than the minimum number of SNPs specified (in the reference data) or if the level of LD across the breakpoint is too high.

The European panel of 1,000 Genomes (phase 3)¹⁶ was used to compute this partitioning, filtering out SNPs with a MAF lower than 1%. The minimum number of SNPs per block was set to 2,500, resulting in 2,495 blocks of, on average, about 1Mb in size. Note that although the method aims to minimize the LD between blocks, some level of LD will generally still exist between adjacent blocks, and as such, they cannot be considered fully independent of each other.

Focusing on the second analysis (Fig. 2)

- In this analysis local genetic sex differences were interrogated using LAVA. However, it was not clear whether the phenotypes being studied were genetically correlated to each other and therefore whether sex differences were therefore expected to be found repetitively. I think it would be helpful to clarify this.

We have added two supplementary correlograms to show the cross-trait global genetic correlations for both males and females. We find some loci to have significant local correlations for several traits (mostly metabolic and weight-related traits). However, these are the exceptions, and most loci are only significant for 1-2 traits. See the added text in the manuscript (page 7):

We note that 8% (942 / 11,935) and 7% (842 / 12,246) of cross-trait pairs have significant Bonferroni-corrected global genetic correlations ($r_{g, global}$) for females and males, respectively (see Supplementary Figures 5 & 6). These mostly cluster within weight- and fat-distribution-related phenotypes, and some loci with significant $r_{g, local}$ between males and females are expected to repeat across these phenotypes. Indeed, locus 2310 (19:3085447:3893909) has a significant $r_{g, local}$ for 14 metabolic traits. However, this is not the norm, and 80% of loci with at least one significant $r_{g, local}$ are found for at most two traits.

- Included in this analysis were all traits and locus pairs with $h^2_{local} P < 1e-04$. However, it is not clear why this P-value cut-off was used. There is no mention of this in the main text, nor in the methods where it is also mentioned (page 13). Could this be explained?

This filtering step removes regions with little genetic variance that are not worth further analysis. Genetic correlations cannot exist if there is no genetic variance to begin with. Applying a threshold, such as $1.00e-4$, improves computational efficiency while reducing the multiple testing burden.

We have added clarification to the methods (page 21):

All LAVA analyses were performed on 2495 semi-linkage disequilibrium independent blocks of approximately 1Mb in size (Data availability). Local genetic correlations were only estimated for loci with sufficient evidence of heritability in both males and females. This is because a genetic correlation cannot exist if no genetic variance is present, and filtering loci with little to no heritability improves computational efficiency while reducing the multiple testing burden. To this end, we applied a h^2_{local} threshold of $p < 1.00e-04$. Using this threshold reduced the number of loci by 93% while retaining 82% of loci with genome-wide significant SNPs in both males and females (see Supplementary Table 3).

- Computed local genetic correlations (r_{glocal}) for males and females were generated and a test run for deviation from one, presumably perfect correlation. It would be helpful if this was explained and stated for the reader.

We have added this to the manuscript (page 7):

To test for local genetic sex differences, we computed local genetic correlations ($r_{\text{g, local}}$) with LAVA for every trait and locus with sufficient genetic signal (i.e., we used a h^2_{local} p-value threshold of $p < 1.00e-04$; see Methods) for both males and females ($N_{\text{loci}} = 11259$) and tested if the $r_{\text{g, local}}$ is significantly different from one (i.e. we tested for deviation from perfect correlation) at a Bonferroni-corrected significance threshold of $p < 0.05 / 11259 = 4.44e-06$.

Focusing on the third analysis (Fig. 3 + Fig. 4)

- Next, to test that LAVA was producing valid results, computed global genetic correlations (r_{gglobal}) using LDSC were generated and compared. While finding good agreement between methods is helpful, given that the major innovation of this analysis is local heritability analyses the emphasis should be on traits with discrepancies. Consequently, Figure 4 is important in that it visualises the exact traits and genomic locations which have LAVA-LDSC discrepancies. For individual discrepancy examples mentioned in the text (page 6), it would be helpful to explicitly label them in figure 4.

We agree that highlighting the loci in Figure 4, which we also discuss in the text, would be helpful. As such, we have labeled the loci we discuss on page 9 in Figure 4.

Fourth analysis (Fig. 5)

- To further understand why r_{gglobal} and the weighted mean of r_{glocal} generated different magnitudes, they extended LAVA to test for the equality of genetic effects. This incorporates a test for equality of the direction of genetic effects and their relative magnitudes (i.e. correlations), but also for equality of their absolute magnitude. I thought this was a very interesting new metric. Is this functionality now formalised in the LAVA package and could that be stated?

While it is not yet available in the current release of the LAVA R package, the R code to run these tests can be downloaded from the Zenodo link in the manuscript. Please see:

<https://doi.org/10.5281/zenodo.12772889>

Fifth analysis (Fig. 6)

- I thought use of testosterone as a positive control for the biological validation of the equality test was very valuable. In this context, it would also be helpful to highlight an example which is a positive control for females?

We chose testosterone as a positive control because it is a clear phenotype in terms of known underlying biology and has clearly different underlying biology for males and females—we are glad the reviewer appreciates this choice. Unfortunately, we are not aware of other traits for which the underlying biology is known and clearly different between males and females, so we did not add another positive control.

Furthermore, power differences across the sexes are touched upon, but given the importance I wonder if this could be explored further and modelled.

We have now added simulations to assess the effect of GWAS sample size differences between the sexes. While we did find power asymmetries at extreme sample size differences between males and females (sample size ratio ≥ 2), we did not find any notable asymmetries at sample size ratios typical of the traits we study (median sample size ratio = 1.16). Nonetheless, we note in the manuscript that when sample sizes start to diverge more strongly, there is more power to detect heritability differences when the sex with the smaller sample size has the larger heritability. See Supplementary Figure 3 and page 5:

Approximately 62% of all significant h^2_{local} s are significant in only one of the sexes. Across all traits, we identify more loci with significant h^2_{local} s in females than males (10614 loci across 153 traits vs. 8767 loci across 151 traits). This is at least partly due to the sex-stratified GWASs' sample sizes being mostly larger in females, increasing the power of h^2_{local} analyses in females relative to males. Specifically, 153 out of 157 traits have larger sample sizes in females with a median female:male sample size ratio of 1.16 (see Supplementary Figure 2 and Supplementary Table 1 for exact sample sizes). This is a consequence of the female sampling bias in the UK Biobank. Simulations show that this can result in power asymmetries at large sample size differences (see Supplementary Figure 3 and Methods), with somewhat greater power to detect differences in heritability if the larger heritability is for the sex with the smaller sample rather than the other way around. However, this power asymmetry is largely negligible for the sample size differences typical of the UK Biobank.

Minor comments

Page 12: '(URLs)' in text rather than actual script repository link

Done

Page 13: '(Data availability)' in text rather than link

Done

Page 6: 'very close one' should be 'very close to one'

It was fixed as suggested.

References

1. Schipper, M. *et al.* Gene prioritization in GWAS loci using multimodal evidence. 2023.12.23.23300360 Preprint at <https://doi.org/10.1101/2023.12.23.23300360> (2024).
2. Coviello, A. D. *et al.* A genome-wide association meta-analysis of circulating sex hormone-binding globulin reveals multiple Loci implicated in sex steroid hormone regulation. *PLoS Genet.* **8**, e1002805 (2012).
3. Sinnott-Armstrong, N., Naqvi, S., Rivas, M. & Pritchard, J. K. GWAS of three molecular traits highlights core genes and pathways alongside a highly polygenic background. *eLife* **10**, e58615 (2021).
4. Flynn, E. *et al.* Sex-specific genetic effects across biomarkers. *Eur. J. Hum. Genet.* **29**, 154–163 (2021).
5. Yang, M.-L. *et al.* Sex-specific genetic architecture of blood pressure. *Nat. Med.* 1–11 (2024) doi:10.1038/s41591-024-02858-2.
6. Werme, J., van der Sluis, S., Posthuma, D. & de Leeuw, C. A. An integrated framework for local genetic correlation analysis. *Nat. Genet.* **54**, 274–282 (2022).
7. Boraska, V. *et al.* Genome-wide meta-analysis of common variant differences between men and women. *Hum. Mol. Genet.* **21**, 4805–4815 (2012).
8. Pirastu, N. *et al.* Genetic analyses identify widespread sex-differential participation bias. *Nat. Genet.* **53**, 663–671 (2021).
9. Benonisdottir, S. & Kong, A. Studying the genetics of participation using footprints left on the ascertained genotypes. *Nat. Genet.* **55**, 1413–1420 (2023).
10. Zhu, C. *et al.* Amplification is the primary mode of gene-by-sex interaction in complex human traits. *Cell Genomics* 100297 (2023) doi:10.1016/j.xgen.2023.100297.
11. Ferretti, M. T. *et al.* Sex differences in Alzheimer disease — the gateway to precision medicine. *Nat. Rev. Neurol.* **14**, 457–469 (2018).

12. Fu, J. *et al.* Effects of Sex on the Relationship Between Apolipoprotein E Gene and Serum Lipid Profiles in Alzheimer's Disease. *Front. Aging Neurosci.* **14**, (2022).
13. Riedel, B. C., Thompson, P. M. & Brinton, R. D. Age, APOE and sex: Triad of risk of Alzheimer's disease. *J. Steroid Biochem. Mol. Biol.* **160**, 134–147 (2016).
14. Altmann, A., Tian, L., Henderson, V. W., Greicius, M. D. & Investigators, A. D. N. I. Sex modifies the APOE-related risk of developing Alzheimer disease. *Ann. Neurol.* **75**, 563–573 (2014).
15. Dunk, M. M. *et al.* Associations of dietary cholesterol and fat, blood lipids, and risk for dementia in older women vary by APOE genotype. *Alzheimers Dement.* **19**, 5742–5754 (2023).
16. Auton, A. *et al.* A global reference for human genetic variation. *Nature* **526**, 68–74 (2015).

REVIEWER COMMENTS

We thank the reviewers for their efforts. Below we provide a point-by-point response to the comments and suggestions.

Reviewer #1 (Remarks to the Author):

The revision includes additional analysis using the gene patronization approach FLAMES to further annotate regions of interest, enhancing the manuscript's rigor. However, the performance of this approach in identifying causal genes needs clarification. For instance, how many genes were identified per region? Do the number of genes and their effects differ between sexes? Quantitative illustrations would aid in interpreting this analysis.

We thank the reviewer for pointing out this lack of detail: FLAMES prioritizes exactly one gene per region tested – we have now clarified this in the manuscript (see Methods, p. 30).

We ran FLAMES not for all 157 traits but only for testosterone, diastolic blood pressure, and LDL levels, as those traits we examined more closely; this is also stated more explicitly now in the manuscript (see page 14).

In addition, we now provide more detail on how many genomic risk loci overlapped between males and females and how often the same gene was predicted to be causal for overlapping genomic risk loci for each of these traits:

Testosterone (p. 14):

Most testosterone GWAS hits did not overlap between males and females¹⁶: Out of 47 loci in the female GWAS and 112 loci in the male GWAS, four loci overlapped. Differences may also exist for overlapping loci. For one of the overlapping loci, we found AKR1C3, a gene associated with testosterone synthesis and metabolism, to map to a GWAS hit in males (10:4987497:5487497) and females (10:4820686:5320686). However, the lead SNPs differ and are on opposite sides of AKR1C3, and both equality_{raw} ($p = 5.19e-12$) and equality_{std} ($p = 1.52e-14$) showed significant differences for this gene. There was no difference in h^2_{local} ($p = 0.24$), but the 1Mb locus that contained AKR1C3 had a $r_{g,local}$ of 0.18, which was significantly different from 1 ($p < 1e-8$) but not from 0 ($p = 0.17$). As such, while AKR1C3 contains signals in both males and females, the pattern of SNP associations is markedly different. Moreover, the gene prioritization tool FLAMES²³ predicted different causal genes at this locus, namely AKR1C2 for females and AKR1C4 for males, which we also found to be significantly different in the equality test (see Figure 6). FLAMES predicted the same causal gene for the other three overlapping loci.

Diastolic blood pressure (p. 14):

Furthermore, out of 101 female GWAS loci and 45 male GWAS loci, 26 loci overlapped. Of these 26 overlapping loci, 22 were predicted to have the same causal gene, while 4 were predicted to have different causal genes between males and females. However, none of these discordant genes were significant in the equality test.

Low-density lipoprotein (LDL) cholesterol levels (p. 15):

Lastly, out of 85 female GWAS loci and 56 male GWAS loci for LDL-direct levels, we found 32 that overlapped between both. FLAMES predicted the same causal gene for 25 of these, and different causal genes for the other seven. However, only SUGP1 significantly differed between males and females in $equality_{raw}$ ($p = 8.22e-13$) and $equality_{std}$ ($p = 9.62e-13$). Interestingly, SUGP1 has previously been shown to be associated with coronary artery disease and cholesterol metabolism¹.

To facilitate biological discovery, validating key findings with independent cohorts is crucial to extend the study's impact. Replicating top hits is necessary to confirm the existence of sex differential genetic effects. The lack of publicly available sex-stratified GWAS data limits replication. I suggest exploring resources like the Japan Biobank, which has sex-stratified GWAS, or other biobank studies.

We agree that validating our main findings in independent cohorts would be very beneficial. As per suggestion, we explored BioBank Japan. There are indeed a number of phenotypes for which sex-stratified GWAS summary statistics are publicly available (see <http://jenger.riken.jp/en/result>). However, this only applies to binary disease phenotypes, which have not (and cannot currently) been included in our methods. We have contacted BioBank Japan to see whether we missed more phenotypes, but they verified that no sex-stratified summary statistics for quantitative phenotypes have been generated. We then attempted to find other sources of sex-stratified GWAS summary statistics of quantitative phenotypes of sufficient size that exclude the UK Biobank, including literature and GWAS catalog searches. The only source that qualified was the GIANT consortium, which released sex-stratified GWAS summary statistics of BMI, Waist-Circumference, and Hip-Circumference without the UK Biobank in 2015 (all subsequent GWASs either included the UK Biobank or did not release sex-stratified ones). While the sample size of the GIANT GWAS of BMI was relatively large (mean N females: 129,807 and mean N males: 103,880), it was considerably smaller than the UKB (mean N females: 193,570 and mean N males: 166,413), it had less global genetic signal (h_{global}^2 : 17% vs. 25% for females and 17% vs. 26% for males), and identified many fewer risk loci (39 vs. 159 for females and 33 vs. 149 for males). Similarly, we identified only nine loci that had sufficient genetic signal (with $p < 1e-4$; threshold applied in all our $r_{g,local}$ analyses) in both males and females to compute $r_{g,local}$ (compared to 212 loci in the UKB). Of these nine loci, seven had $r_{g,local}$ close to one, while two $r_{g,local}$ were out of bounds. The same loci were also not significantly different from 1 in the UKB. There is even less signal for Waist-Circumference and Hip-Circumference. As such, due to this lack of power and genetic signal in the GIANT summary statistics, we decided not to include this in our manuscript, as we deemed an underpowered study not relevant for replication. If the reviewer knows of any other source we can use for a replication analysis, we are eager to hear about them and will happily include this.

In the manuscript, we now emphasize the need for replication in the limitations and urge GWAS authors to release sex-stratified summary statistics in all future analyses (see p. 20):

Third, we only considered data from the UK Biobank. This is because it is the only large biobank we know for which sex-stratified GWAS summary statistics for quantitative traits have been computed and made publicly available for hundreds of traits. We urge the authors of future GWASs to release sex-stratified summary statistics.

While identifying potentially causal genes is beneficial, their biological relevance should be discussed with consideration of sex differences. The impact of candidate genes on phenotypic variance between sexes hasn't been comprehensively addressed in the main text. Additionally, I recommend releasing the data from this new analysis to prioritize gene candidates.

We thank the reviewer for pointing us to this lack of detail. We have added a paragraph to the Discussion about the role of putative causal genes between the sexes (p. 19):

On a variant level, it is expected that the identity of causal variants does not differ between the sexes and that genetic sex differences are due to causal variant effect sizes only. On a gene level, we would therefore expect the same, namely that the same genes are causal and that merely the h_{local}^2 per gene may differ. However, there may be several scenarios that can give rise to diverging causal gene predictions. First, if GxS interactions are sufficiently strong, the effect of genes may start to appear qualitatively different. This seems to be the case for Testosterone, which appears to be two separate phenotypes altogether in males and females with few overlapping risk loci, global genetic correlations of zero, and many genes significantly different in the equality test. Second, different causal genes may be predicted when a locus contains more than one causal gene, but the relative importance differs by sex, such that the top-ranking genes are not the same. Third, a locus may only reach genome-wide significance in one sex and may thus not yield a predicted causal gene in our analysis with FLAMES for the other sex (such as COL4A1 for diastolic blood pressure). Lastly, FLAMES selects the top-ranking gene as the most likely causal one, while the second- and third-ranking genes may be only slightly less likely. This may explain why most predicted causal genes that differed between males and females were not significantly different in the equality test. Moreover, FLAMES is expected to have 75% precision and, as such, some uncertainty in the predictions is expected.

As per request, we made the $r_{g,local}$ h_{local}^2 equality test results (for the 1Mb regions and genes) and plots for all 157 traits, as well as FLAMES gene predictions for testosterone, diastolic blood pressure, and LDL levels, available for download from Zenodo (<https://doi.org/10.5281/zenodo.15213372>). In the manuscript, we have clarified which results exactly are available for download in the “Data availability” subsection of the manuscript (p. 31):

Sex-stratified GWAS summary statistics:
https://github.com/Nealelab/UK_Biobank_GWAS
LAVA locus definition file: <https://github.com/josefin-werme/LAVA>
1000 Genomes LD reference file for LAVA: <https://ctg.cncr.nl/software/lava>
Scripts, plots, and results for all LAVA analyses and all 157 traits, and FLAMES results for Testosterone, Diastolic blood pressure, and LDL direct:
<https://doi.org/10.5281/zenodo.15213372>

The author presents the equality test, offering intriguing insights into sex differences in genetics. To enhance understanding, it would be beneficial to provide more biological interpretation and examples of how the proposed model can elucidate these differences.

The benefit of the equality test is to go beyond testing for equality of the direction of genetic effects and their relative magnitudes (i.e., correlations) but also to consider their absolute and standardized magnitudes. We find 45 1Mb loci across 23 phenotypes where the $r_{g,local}$ s are close to one or not significantly different from one, but where the equality test is significant on at least one scale after Bonferroni correction. Had we only considered $r_{g,local}$, we would have missed these differences. We have added this quantification to the Results (p. 12).

One locus in particular (locus 2351 19:45040933:45893307) is significant in the equality test for five phenotypes, while the $r_{g,local}$ s are close to one. The phenotypes are all lipid-related, namely Apolipoprotein B, Total Cholesterol, HDL Cholesterol, LDL Cholesterol, and Triglycerides. The gene-level equality test shows that APOE, specifically, is significantly different between males and females for all these phenotypes. For most of these (except Triglycerides), the equality test is highly significant on both scales, with larger h^2_{local} in females, particularly for LDL Cholesterol. This means that the APOE SNP effects have larger effects on these lipid biomarkers both in absolute and relative terms in females. Again, these effects would have been missed by only considering $r_{g,local}$ s.

We have added analyses for these additional phenotypes to the Results (p. 15), added Supplementary Figure 7 displaying the local heritabilities of the lipid traits, and extended the Discussion to highlight the potential implications of these results for Cardiovascular Disease and Alzheimer's Disease (p. 19):

[...] we found APOE to have different magnitudes of genetic effects on the raw and standardized scales for several lipid-related phenotypes while the $r_{g,local}$ was not different from one, leading to h^2_{local} estimates that are up to twice as large in females. As such, APOE must interact with male or female hormones or with external environmental variables (e.g., diet) to result in these divergent genetic effects. Future studies could compare whether these differences mediate sex-dimorphic effects of APOE on Alzheimer's Disease²⁹⁻³³ and Cardiovascular Disease^{34,35} risk.

Reviewer #2 (Remarks to the Author):

Thanks to the authors for addressing my comments!

I have one follow-up question: Given the female sampling bias in the UK Biobank and the importance of validating findings in a replication cohort, can the authors try to replicate the loci found in UKBB with sex differences in another biobank?

Reviewer 1 has a similar comment, to which we have responded in detail. We would be eager to add this, but such sex-stratified summary statistics of sufficient size that exclude the UK Biobank are not (yet) available. We copy the same reply here:

We agree that validating our main findings in independent cohorts would be very beneficial. As per suggestion, we explored BioBank Japan. There are indeed a number of phenotypes for which sex-stratified GWAS summary statistics are publicly available (see <http://jenger.riken.jp/en/result>). However, this only applies to binary disease phenotypes, which have not (and cannot currently) been included in our methods. We have contacted BioBank Japan to see whether we missed more phenotypes, but they verified that no sex-stratified summary statistics for quantitative phenotypes have been generated.

We then attempted to find other sources of sex-stratified GWAS summary statistics of quantitative phenotypes of sufficient size that exclude the UK Biobank, including literature and GWAS catalog searches. The only source that qualified was the GIANT consortium, which released sex-stratified GWAS summary statistics of BMI, Waist-Circumference, and Hip-Circumference without the UK Biobank in 2015 (all subsequent GWASs either included the UK Biobank or did not release sex-stratified ones). While the sample size of the GIANT GWAS of BMI was relatively large (mean N females: 129,807 and mean N males: 103,880), it was considerably smaller than the UKB (mean N females: 193,570 and mean N males: 166,413), it had less global genetic signal (h_{global}^2 : 17% vs. 25% for females and 17% vs. 26% for males), and identified many fewer risk loci (39 vs. 159 for females and 33 vs. 149 for males). Similarly, we identified only nine loci that had sufficient genetic signal (with $p < 1e-4$; threshold applied in all our $r_{g, local}$ analyses) in both males and females to compute $r_{g, local}$ (compared to 212 loci in the UKB). Of these nine loci, seven had $r_{g, local}$ close to one, while two $r_{g, local}$ were out of bounds. The same loci were also not significantly different from 1 in the UKB. There is even less signal for Waist-Circumference and Hip-Circumference. As such, due to this lack of power and genetic signal in the GIANT summary statistics, we decided not to include this in our manuscript, as we deemed an underpowered study not relevant for replication. If the reviewer knows of any other source we can use for a replication analysis, we are eager to hear about them and will happily include this.

In the manuscript, we now emphasize the need for replication in the limitations and urge GWAS authors to release sex-stratified summary statistics in all future analyses (see p. 20):

Third, we only considered data from the UK Biobank. This is because it is the only large biobank we know for which sex-stratified GWAS summary statistics for quantitative traits have been computed and made publicly available for hundreds of traits. We urge the authors of future GWASs to release sex-stratified summary statistics.

Reviewer #3 (Remarks to the Author):

The authors have addressed the comments appropriately. More specifically, I think the additional text and simulation analyses have improved the readability and value of the manuscript. I still think that the identification of a positive control for females should be possible, and though I appreciate that it is harder, I think more could be done to identify this and run appropriate additional analyses. Nonetheless, overall, I think this manuscript does indeed add to the important field of sex

differences in genetic traits and I would recommend it for publication.

We thank the reviewer for the suggestion to add a positive control for females. Addressing this suggestion, we inspected Oestradiol, the major female sex hormone and most obvious female equivalent to Testosterone. However, the sex-stratified GWASs of Oestradiol contain very little genetic signal, so we refrained from doing an in-depth analysis of Oestradiol. We added text highlighting the lack of genetic signal for Oestradiol and Testosterone's unique genetic architecture in males and females to the Results (p. 14):

The major female sex hormone, Oestradiol, has a very different genetic architecture from the major male hormone, Testosterone. The sex-stratified GWASs of Oestradiol contain little genetic signal, each identifying only one genome-wide significant risk locus and with h_{global}^2 estimates of $\sim 2\%$. Moreover, the $r_{g, global}$ was not significantly different from one.

And to the Discussion (p. 19):

Testosterone [...] appears to be two separate phenotypes altogether in males and females with few overlapping risk loci, global genetic correlations of zero, and many genes significantly different in the equality test.